# BAYESIAN PSEUDO-CORESETS VIA CONTRASTIVE DIVERGENCE

## ABSTRACT

Bayesian methods provide an elegant framework for estimating parameter posteriors and quantification of uncertainty associated with probabilistic models. However, they often suffer from slow inference times, rendering them impractical for scalable applications. To address this challenge, Bayesian Pseudo-Coresets (BPC) have emerged as a promising solution. BPC methods aim to create a small synthetic dataset, known as pseudo-coresets, that approximates the posterior inference achieved with the original dataset. This approximation is achieved by optimizing a divergence measure between the true posterior and the pseudo-coreset posterior. Various divergence measures have been proposed for constructing pseudo-coresets, with forward Kullback-Leibler (KL) divergence being the most successful. However, using forward KL divergence necessitates sampling from the pseudo-coreset posterior, often accomplished through approximate Gaussian variational distributions. Alternatively, one could employ Markov Chain Monte Carlo (MCMC) methods for sampling, but this becomes challenging in high-dimensional parameter spaces due to slow mixing. In this study, we introduce a novel approach for constructing pseudo-coresets by utilizing contrastive divergence. Importantly, optimizing contrastive divergence eliminates the need for approximations in the pseudo-coreset construction process. Furthermore, it enables the use of finite-step MCMC methods, alleviating the requirement for extensive mixing to reach a stationary distribution. To validate our method's effectiveness, we conduct extensive experiments on multiple datasets, demonstrating its superiority over existing BPC techniques. Our implementation is available at https://anonymous.4open.science/r/BPC-CD-E762

## 1 INTRODUCTION

In recent years, contemporary deep learning models have demonstrated exceptional effectiveness in a wide array of applications, spanning computer vision, natural language processing, and speech analysis (Krizhevsky et al., 2017b; Devlin et al., 2018; Amodei et al., 2016; He et al., 2016a; Dosovitskiy et al., 2020; Radford et al., 2021). Conventional deep learning methods rely on one-time training of models providing point estimates (Szegedy et al., 2013). These point estimates are prone to overfitting and often provide overconfident or under-confident outputs (Gawlikowski et al., 2023; Kabir et al., 2018). This prohibits the use of deep learning models in critical applications such as medical, finance, etc (Ker et al., 2017; Cavalcante et al., 2016). Bayesian methods furnish a systematic framework for parameter estimation and quantification of associated uncertainty. Bayesian inference entails sampling from parameter posterior distributions using Markov Chain Monte Carlo (MCMC) techniques (Robert et al., 1999; Robert & Casella, 2011). However, conducting inference based on parameter posterior conditioned on the entire dataset is computationally demanding, particularly as the dataset size, denoted by $N$, increases. The computational complexity of MCMC methods scales with $N$ as $\Theta(NS)$, where $S$ denotes the number of samples (Campbell & Broderick, 2018). This complexity becomes prohibitively high for large $N$. To mitigate this, one often resorts to using a random subset of $M \ll N$ data points for likelihood computation at each iteration (Bardenet et al., 2017; Korattikara et al., 2014; Maclaurin & Adams, 2014; Welling & Teh, 2011; Ahn et al., 2012; Bierkens et al., 2019; Pollock et al., 2020). However, such approximations introduce errors and lead to slow mixing of Markov chains (Johndrow et al., 2020; Nagapetyan et al., 2017; Betancourt, 2015).

Bayesian coresets (Huggins et al., 2016) were introduced to solve the aforementioned problem. Particularly, Huggins et al. (2016) proposed to select a subset of original dataset (also called a coreset) that uniformly approximates the log-likelihood of the original dataset. These coresets are significantly smaller in size than the original dataset, leading to vastly improved sampling efficiency. Further, Campbell & Beronov (2019) proposed to identify the coreset by minimizing Kullback-Leibler (KL) divergence between full data posterior and coreset posterior. However, most of such methods do not scale with data dimension (Manousakas et al., 2020). Particularly, the KL-divergence between the (optimal) coreset posterior and true posterior increases with data dimension. Meaning that for large data dimension, even with the optimal coresets, the KL-divergence is far from optima (zero), implying that the true posterior is not approximated correctly. However, recently few methods have tried to overcome the scalability issue by using lightweight coresets (Bachem et al., 2018).

The Bayesian Pseudo-Coreset (BPC) (Manousakas et al., 2020) approach, as a distinct category of methods, has been proposed to *synthesise* a smaller dataset from the original one, as opposed to selecting a subset, which is the case with coreset methods. The fundamental idea of BPC involves solving an optimization problem over the data space, leading to creation of a 'pseudo' dataset that appropriately approximates the true posterior. The said optimization problem pertains to minimization of a divergence metric between the true posterior and pseudo-coreset posterior. Such a framework removes the constraint for the pseudo-coresets to be a subset of the original dataset. This additional degree of freedom aids in better optimization of the divergence measure. Further, in addition to better approximation of true posterior, pseudo-coreset also come with privacy benefits. Specifically, since pseudo-coresets are not part of original dataset, one can outsource these pseudo-coreset without revealing the original dataset for an user to run inference. Manousakas et al. (2020) also provided theoretical guarantees showing pseudo-coresets are differentially private.

Recently, Kim et al. (2022a) analyzed the BPC construction under different divergence measures such as reverse-KL and Wasserstein divergence. Their analysis revealed that BPC methods under different divergence measures are equivalent to their non-bayesian counterparts. These non-bayesian frameworks are often referred to as 'Dataset Condensation' or 'Dataset Distillation' (Zhao et al., 2021; Zhao & Bilen, 2023; Cazenavette et al., 2022; Wang et al., 2022; Nguyen et al., 2021). In particular, Kim et al. (2022a) showed that minimization of reverse-KL is equivalent to gradient matching (Zhao et al., 2021) and minimization of wasserstein measure is equivalent to matching training trajectory (Cazenavette et al., 2022). They also proposed to use forward-KL for better pseudo-coreset construction due to its ability to capture the support of the distribution, contrasting with reverse-KL, which tends to focus on the distribution's modes. However, computing the gradient of forward-KL requires sampling from the intractable pseudo-coreset posterior. While this can be achieved using MCMC methods, the extensive mixing time of MCMC in high-dimensional parameter spaces renders this approach impractical. As a remedy, Gaussian variational approximation around SGD solutions was employed to simplify and expedite the sampling process. However, the quality of such approximation is unknown and remains a matter of concern.

In our current work, we propose a novel approach: using contrastive divergence instead of forward-KL divergence for pseudo-coreset learning. This has two advantages: (1) It eliminates the need for approximating the pseudo-coreset posterior, enabling the straightforward use of MCMC methods, (2) The Markov chain used in this approach does not require extensive mixing to reach a stationary distribution; only a finite number of steps is needed. These advantages effectively address the challenges associated with using forward-KL divergence. Furthermore, our rigorous experiments demonstrate that our proposed method significantly outperforms previous state-of-the-art BPC methods, thereby confirming that the pseudo-coreset posterior using contrastive divergence better approximates the true posterior. Our contributions can be summarized as follows:

- We propose a new framework for the construction of Bayesian Pseudo-Coreset using contrastive divergence.

- The proposed method avoids any approximation of pseudo-coreset posterior and facilitates the use of finite step MCMC methods during learning phase.

- Extensive experimentation reveals that our method surpasses state-of-the-art BPC methods by substantial margins, affirming the better approximation of the true posterior using contrastive divergence.

## 2 RELATED WORK

### 2.1 BAYESIAN INFERENCE AND OPTIMIZATION

The objective of Bayesian methods is the model the parameter posterior distribution of a probabilistic model. However, apart from some simple models, the exact posterior distributions are generally intractable (Campbell & Broderick, 2019). In such scenarios, one often relies on inference techniques like MCMC methods (Robert et al., 1999; Robert & Casella, 2011) and Variational Inference (VI) (Jordan et al., 1998; Wainwright et al., 2008). Historically, these inference techniques require model-specific tuning based on the path-length parameters, step size (Neal et al., 2011a), and the choice of the variational families (Jaakkola & Jordan, 1997; Jordan et al., 1999). Recent methods (Ranganath et al., 2014; Kucukelbir et al., 2017; Hoffman et al., 2014) have circumvented these issues by introducing a black box approach that requires only basic specifications about the model. For instance, the traditional variational inference methods (Jaakkola & Jordan, 1997; Jordan et al., 1999) relied on closed form gradients of the model (Ranganath et al., 2014) and an approximate distribution for the posterior of the data. Ranganath et al. (2014); Baydin et al. (2018); Kucukelbir et al. (2017) addressed these issues by employing standard transformation over a multivariate Gaussian distribution and used automatic differentiation techniques to calculate the associated gradients. Similarly, for MCMC methods like Hamiltonian Monte Carlo (HMC) (Neal et al., 2011a) traditional practices involved manually tuning of parameters like step size and path length to achieve accurate posterior estimation. Hoffman et al. (2014) addressed this challenge by automatically estimating both of these parameters.

In many modern applications, these methods are required to scale with the size of the datasets. The standard MCMC algorithms are computationally expensive for large datasets, and the sampling process scales linearly with the data size. Recent works (Bardenet et al., 2017; Korattikara et al., 2014; Maclaurin & Adams, 2014; Welling & Teh, 2011; Ahn et al., 2012; Bierkens et al., 2019; Pollock et al., 2020), have tried to mitigate the computational cost associated with inference models by considering only a random subset of data points during MCMC iterations. One of the initial studies in this direction has been conducted by Welling & Teh (2011) where the authors proposed to use stochastic gradient langevin dynamics (SGLD). This iterative learning algorithm utilizes mini-batches of dataset for Bayesian inference. However, unlike other MCMC methods, their approach often leads to a slow mixing rate. Ahn et al. (2012) addressed this issue by sampling from the Gaussian approximation of posterior for a high mixing rate and mimicking the behavior of SGLD using a pre-conditioner matrix for a slow mixing rate. However, Korattikara et al. (2014); Bardenet et al. (2014) have shown that such a sampling approach often leads to a stationary distribution that can have bounded errors under strong conditions of rapid mixing (Maclaurin & Adams, 2014). In contrast, they proposed a new accept/reject strategy to select a subset of the dataset for Bayesian inference. On a similar line, Maclaurin & Adams (2014) proposed to use a collection of Bernoulli latent variables to select a subset of the dataset for likelihood estimation. Bierkens et al. (2019); Pollock et al. (2020) have further proposed to use a zig-zag process and quasi-stationary distribution along with the subsampling approaches for bayesian inference.

### 2.2 BAYESIAN CORESETS

Bayesian coresets (Huggins et al., 2016; Campbell & Broderick, 2018; Campbell & Beronov, 2019; Campbell & Broderick, 2019; Zhang et al., 2021; Naik et al., 2022; Chen et al., 2022) present an alternative strategy to address aforementioned challenges by selecting a small weighted subset of the original dataset which can closely approximate the posterior of the full dataset (Zhang et al., 2021; Huggins et al., 2016). The idea was introduced in Huggins et al. (2016), where a weighted subset of original data was selected to approximate the log-likelihood of the entire dataset up to some multiplicative error over the parameter space. However, the subset produced by such a technique underestimates the posterior distribution and can result in large approximation errors for some models regardless of the coreset size. Campbell & Broderick (2018) addressed this issue using greedy iterative geodesic ascent (GIGA), that optimally scales the log-likelihood of the coreset to better approximate the entire log-likelihood of the dataset. It further provided a uniform bounded error for all the models. To further enhance the scalability, Campbell & Broderick (2019) tackled the model and data-specific assumptions made in prior work regarding coreset construction. They constructed Bayesian coreset by solving a sparse vector sum based approximation using frank-wolfe (Frank et al., 1956) based

solvers. Recent works (Zhang et al., 2021; Naik et al., 2022; Chen et al., 2022) have focused on improving the speed of coreset construction using accelerated optimization methods, quasi-newton refinement, and sparse-hamiltonian flows. However, since the KL divergence between the posteriors of the optimal coreset and the original dataset increases with the data dimensionality (Manousakas et al., 2020), these methods do not easily scale up in high-dimensions.

## 2.3 BAYESIAN PSEUDO-CORESET

Manousakas et al. (2020) proposed to use a collection of synthetic data to scale the Bayesian inference to high dimensional datasets. Particularly, they frame the problem as divergence minimization between the posteriors associated with the synthetic and the original dataset. The synthetic set generated through this technique is called 'Bayesian Pseudo-Coreset' (BPC). Compared to Bayesian coresets, these methods scale more efficiently with data dimensions and yield a more accurate posterior approximation.

Manousakas et al. (2020) formalized the given problem by minimizing the reverse-KL divergence between the posterior of original data and the posterior of synthetic data. On similar lines, Kim et al. (2022a) demonstrated that other divergence metrics, such as Wasserstein distance and forward-KL divergence, can be used to generate pseudo-coreset. In contrast to reverse-KL, which primarily focuses on the modes of the distributions, forward-KL provides a mechanism to better capture the support of the posterior distribution. To efficiently calculate the forward-KL divergence Kim et al. (2022a) used a Gaussian variational approximation of the posterior distribution. However, the quality of such an approximation and its impact on the overall performance of the pseudo-coreset is unknown. Further, computing the gradient of forward-KL requires sampling from an intractable posterior of pseudo-coreset using MCMC methods, which is not straightforward in practice.

## 2.4 CORESETS AND DATASET CONDENSATION

While Bayesian coreset focuses on selecting data points to facilitate Bayesian inference, coreset selection strategies have been proposed for other algorithms like geometric approximation (Agarwal et al., 2005), mixture models (Feldman et al., 2011), K-means clustering (Feldman & Langberg, 2011; Feldman et al., 2020; Bachem et al., 2016) and DP means (Bachem et al., 2015). Similarly, for deep learning models, Mirzasoleiman et al. (2020); Killamsetty et al. (2021a;b) have introduced subset selection techniques that leverage gradient matching and meta-learning algorithms. Recent works (Welling, 2009; Castro et al., 2018; Rebuffi et al., 2017; Belouadah & Popescu, 2020; Sener & Savarese, 2017; Farahani & Hekmatfar, 2009), have further proposed strategies to choose a representative and diverse set of samples from the original dataset. These methods aim to create a generic subset by removing redundant data points. Herding-based coreset methods (Welling, 2009; Castro et al., 2018; Rebuffi et al., 2017; Belouadah & Popescu, 2020) select such samples by minimizing the distance between the feature centroids of the coreset, and the original dataset. While K-center-based coreset techniques (Sener & Savarese, 2017; Farahani & Hekmatfar, 2009; Guo et al., 2022) pick the most diverse and representative samples by optimizing a submodular function (Farahani & Hekmatfar, 2009). Contrary to K-center and herding-based coreset selection methods, forgetting-based coreset (Toneva et al., 2018) removes the easily forgettable samples from the training dataset.

Rather than selecting a subset of data points from the training set, dataset condensation methods aim to generate a synthetic set that emulates the characteristics of the original dataset. For example, in gradient based dataset condensation techniques (Zhao et al., 2021; Yu et al., 2023; Lee et al., 2022; Jiang et al., 2022) the synthetic samples are generated by aligning the gradients of a model trained using original and synthetic datasets. Similarly, meta-learning based methods (Wang et al., 2018; Deng & Russakovsky, 2022; Nguyen et al., 2021; Loo et al., 2022; Zhou et al., 2022) generate these synthetic samples by matching the validation performance of a model trained using the entire dataset with the performance of a model trained using the synthetic set. Cazenavette et al. (2022); Li et al. (2022); Du et al. (2023) propose generating the synthetic dataset using long-horizon trajectories, ensuring that the models learn similar trajectories during optimization. While distribution matching methods (Zhao & Bilen, 2023; Wang et al., 2022; Zhao & Bilen, 2022; Zhao et al., 2023) generate a condensed synthetic set with a similar feature distribution as the original dataset. Recent works (Liu et al., 2023; Zhang et al., 2023; Cazenavette et al., 2023) have further focused on improving the

performance and computational complexity of existing dataset condensation techniques by using representative samples from the training set, model augmentation techniques, and generative model for learning the synthetic set.

## 3 PROPOSED METHODOLOGY

### 3.1 BAYESIAN PSEUDO-CORESETS

Consider a dataset $(\mathbf{x}, \mathbf{y}) = \{(\mathbf{x}_i, y_i)\}_{i=1}^n$ consisting of $n$ data points. Now consider a synthetic (learnable) dataset $(\tilde{\mathbf{x}}, \tilde{\mathbf{y}}) = \{\tilde{\mathbf{x}}_i, \tilde{y}_i\}_{i=1}^m$ such that $\mathbf{y}$ and $\tilde{\mathbf{y}}$ share the same label space and $m \ll n$. Let, $\theta \in \Theta$ be the parameter of a discriminative / classification model. Then the parameter posteriors corresponding to original and synthetic data, $\pi(\theta|\mathbf{x})$ and $\pi(\theta|\tilde{\mathbf{x}})$ are given by

$$\pi_{\mathbf{x}} \triangleq \pi(\theta|\mathbf{x}) = \frac{\pi_0(\theta)}{Z(\mathbf{x})} \exp\left(\sum_{i=1}^n \log \pi(y_i|\mathbf{x}_i, \theta)\right) \tag{1}$$

$$\pi_{\tilde{\mathbf{x}}} \triangleq \pi(\theta|\tilde{\mathbf{x}}) = \frac{\pi_0(\theta)}{Z(\tilde{\mathbf{x}})} \exp\left(\sum_{i=1}^m \log \pi(\tilde{y}_i|\tilde{\mathbf{x}}_i, \theta)\right) = \frac{\pi_0(\theta)}{Z(\tilde{\mathbf{x}})} \exp\left(-E(\tilde{\mathbf{x}}, \theta)\right) \tag{2}$$

where,

$$Z(\mathbf{x}) = \int_\Theta \pi_0(\theta) \exp\left(\sum_{i=1}^n \log \pi(y_i|\mathbf{x}_i, \theta)\right) d\theta, \quad Z(\tilde{\mathbf{x}}) = \int_\Theta \pi_0(\theta) \exp\left(-E(\tilde{\mathbf{x}}, \theta)\right) d\theta \tag{3}$$

are appropriate normalizing constants. Here, $\pi_0(\theta)$ is the prior distribution and $E(\tilde{\mathbf{x}}, \theta) = -\sum_{i=1}^m \log \pi(\tilde{y}_i|\tilde{\mathbf{x}}_i, \theta)$ is the sum of negative log-likelihoods which can be treated as a generic potential or energy function. Since $n$ is often very large, the posterior estimation using $\pi_{\mathbf{x}}$ is computationally expensive and infeasible. However, an appropriate approximation such as $\pi_{\tilde{\mathbf{x}}}$ where $m \ll n$, allows one to overcome this hurdle. In particular, this approximation is carried out by solving the following optimization problem:

$$\tilde{\mathbf{x}}^* = \arg\min_{\tilde{\mathbf{x}}} \ D\left(\pi_{\mathbf{x}}, \pi_{\tilde{\mathbf{x}}}\right) \tag{4}$$

where, $D(\cdot, \cdot)$ is a divergence measure between two distributions. Recently, Kim et al. (2022a) showed the results for above optimization problem under different divergence metrics. Specifically, they analyzed the results with reverse-KL and wasserstein divergence; consequently drawing equivalence with dataset condensation methods like gradient matching (Zhao et al., 2021) and MTT (Cazenavette et al., 2022). Further, they propose an alternative solution by using forward-KL divergence as it encourages a model to cover the entire target distribution in contrast to reverse-KL which encourages mode capturing models. The gradient of the forward-KL divergence, as derived in Kim et al. (2022a), is expressed as follows:

$$\nabla_{\tilde{\mathbf{x}}} D_{KL}\left(\pi_{\mathbf{x}}||\pi_{\tilde{\mathbf{x}}}\right) = \mathbb{E}_{\pi_{\tilde{\mathbf{x}}}}\left[-\nabla_{\tilde{\mathbf{x}}} E(\tilde{\mathbf{x}}, \theta)\right] + \nabla_{\tilde{\mathbf{x}}} \mathbb{E}_{\pi_{\mathbf{x}}}\left[E(\tilde{\mathbf{x}}, \theta)\right] \tag{5}$$

This gradient computation necessitates the calculation of expectations with respect to the probability distributions $\pi_{\mathbf{x}}$ and $\pi_{\tilde{\mathbf{x}}}$. However, the presence of intractable partition functions ($Z(\mathbf{x})$ and $Z(\tilde{\mathbf{x}})$) poses challenges in efficiently sampling from these posterior distributions. One can resort to MCMC methods such as langevin dynamics or hamiltonian monte-carlo for sampling, however, due to large dimension of $\Theta$-space, the mixing-time of these methods is very large and in-efficient in practice. To overcome this issue, Kim et al. (2022a) employs gaussian variational approximations for these posteriors, rendering the sampling process computationally feasible. Specifically, gaussian distributions are used, centered around parameters obtained from Stochastic Gradient Descent (SGD) trajectories of $\mathbf{x}$ and $\tilde{\mathbf{x}}$ (cf. (Kim et al., 2022a) for details).

In practice, since $m$ (number of samples in pseudo-coreset) is generally very small, the SGD trajectories of $\tilde{\mathbf{x}}$ might overfit, leading to erroneous approximations. Therefore, it is preferable to bypass such approximations and sample directly from the exact posteriors. In this work, we propose to work with contrastive divergence (Hinton, 2002) instead of forward-KL to construct the pseudo-coreset. Specifically, using contrastive divergence leads to a loss objective where $\pi_{\tilde{\mathbf{x}}}$ can

be used as it is without any approximation. The key idea behind this is that instead of minimizing forward-KL, contrastive divergence minimizes difference between two forward-KL terms, that results in cancellation of expectation w.r.t $\pi_{\tilde{\mathbf{x}}}$ allowing us to circumvent this approximation. We describe this in detail in next section.

## 3.2 CONTRASTIVE DIVERGENCE FOR BPC

As mentioned earlier, we propose to work with contrastive divergence instead of forward-KL for construction of pseudo-coresets. The concept of contrastive divergence was initially introduced by seminal work in Hinton (2002).The central premise behind contrastive divergence hinges on a straightforward insight: whereas minimizing forward KL divergence necessitates a term that involves sampling from $\pi_{\tilde{\mathbf{x}}}$, minimizing the difference between two forward KL divergences leads to the nullification of this term. More explicitly, the contrastive divergence is defined as:

$$\mathcal{L}_{CD} = D_{KL}(\pi_{\mathbf{x}}||\pi_{\tilde{\mathbf{x}}}) - D_{KL}(\Pi_E^k \pi_{\mathbf{x}}||\pi_{\tilde{\mathbf{x}}}) \tag{6}$$

where, $\Pi_E^k(\cdot)$ is an MCMC transition kernel for $\pi_{\tilde{\mathbf{x}}}$ and $\Pi_E^k \pi_{\mathbf{x}}$ represents $k$ sequential MCMC transitions starting from $\pi_{\mathbf{x}}$. For brevity, let us denote $\bar{\pi}_{\mathbf{x}}$ as $\Pi_E^k \pi_{\mathbf{x}}$. As shown in Hinton (2002), the gradient of the above objective is approximately given by:

$$\nabla_{\tilde{\mathbf{x}}}\mathcal{L}_{CD} = \mathbb{E}_{\pi_{\mathbf{x}}}\left[\nabla_{\tilde{\mathbf{x}}}E(\tilde{\mathbf{x}},\theta)\right] - \mathbb{E}_{\bar{\pi}_{\mathbf{x}}}\left[\nabla_{\tilde{\mathbf{x}}}E(\tilde{\mathbf{x}},\theta)\right] \tag{7}$$

It is worth noting that the gradient estimation in the above equation does not necessitate sampling from $\pi_{\tilde{\mathbf{x}}}$. Instead, it calls for sampling from $\pi_{\mathbf{x}}$ and $\bar{\pi}_{\mathbf{x}}$. In this context, we can employ a variational posterior to approximate $\pi_{\mathbf{x}}$ and use MCMC sampling techniques (e.g. langevin dynamics (Bohdal et al., 2020)) starting from $\pi_{\mathbf{x}}$ to sample from $\bar{\pi}_{\mathbf{x}}$. Notably, unlike in Eq. 5, the MCMC sampling utilized here only needs to run for finite $k$ steps, alleviating the requirement for substantial Markov chain mixing.

In particular, we use gaussian variational posterior ($q_{\mathbf{x}}$) to approximate $\pi_{\mathbf{x}}$. Then, a $k$-step MCMC starting from $q_{\mathbf{x}}$ should be used as a variational substitute for $\bar{\pi}_{\mathbf{x}}$:

$$q_{\mathbf{x}}(\theta) = \mathcal{N}(\theta; \theta_{\mathbf{x}}, \Sigma_{\mathbf{x}}), \quad \bar{q}_{\mathbf{x}}(\theta) = \Pi_E^k\, q_{\mathbf{x}}(\theta) \tag{8}$$

where, $\theta_{\mathbf{x}}$ is the MAP solution computed for $\mathbf{x}$. Here, one can note that making an approximation for $\pi_{\mathbf{x}}$ is enough unlike previous methods where additional approximations for $\pi_{\tilde{\mathbf{x}}}$ is also required. Hence, the final gradient estimate is obtained as

$$\nabla_{\tilde{\mathbf{x}}}\mathcal{L}_{CD} \approx \mathbb{E}_{q_{\mathbf{x}}}\left[\nabla_{\tilde{\mathbf{x}}}E(\tilde{\mathbf{x}},\theta)\right] - \mathbb{E}_{\bar{q}_{\mathbf{x}}}\left[\nabla_{\tilde{\mathbf{x}}}E(\tilde{\mathbf{x}},\theta)\right] \tag{9}$$

$$\approx \nabla_{\tilde{\mathbf{x}}}\frac{1}{N}\sum_{j=1}^{N}\left[E\left(\tilde{\mathbf{x}}, \theta_{\mathbf{x}} + \Sigma_{\mathbf{x}}^{1/2}\varepsilon_{\mathbf{x}}^{(j)}\right) - E\left(\tilde{\mathbf{x}}, \mathtt{sg}\left(\bar{\theta}^{(j)}\right)\right)\right] \tag{10}$$

where, $\mathtt{sg}(\cdot)$ denotes stop-gradient operator, $\varepsilon_{\mathbf{x}}^{(j)} \sim \mathcal{N}(0,I)$ and $\bar{\theta}^{(j)}$ is obtained via running $k$-step MCMC starting from $\left(\theta_{\mathbf{x}} + \Sigma_{\mathbf{x}}^{1/2}\varepsilon_{\mathbf{x}}^{(j)}\right)$.

For computational efficiency, we assess the parameter posterior with $\mathbf{x}$ using expert trajectories similar to Kim et al. (2022a). In essence, expert trajectories represent sequences of parameters obtained while training a model on the dataset $(\mathbf{x}, \mathbf{y})$. Each of these sequences is termed as 'parameter trajectory,' and the collection of these trajectories, acquired through various training instances, is known as 'expert trajectories.' This eliminates the need to compute MAP solutions for $\mathbf{x}$ ($\theta_{\mathbf{x}}$) at each training step. During training, we randomly pick a parameter from these trajectories to calculate the objective function.

## 4 EXPERIMENTS AND RESULTS

### 4.1 EVALUATION DETAILS

We evaluate our method both quantitatively and qualitatively on several BPC-benchmark datasets with different compression ratios, i.e., the number of images generated per class (ipc). In particular, we perform our experiments on six different datasets, namely, CIFAR10 (Krizhevsky & Hinton,

Table 1: Comparison of the proposed method with BPC baselines. The results are noted in the form of (mean $\pm$ std. dev) where we have obtained test accuracy over five independent runs on the pseudo-coreset. The best performer across all methods is denoted in bold ($\boldsymbol{x \pm s}$).

| | ipc | Ratio(%) | BPC-rKL(sghmc) | | BPC-W (sghmc) | | BPC-fKL (hmc) | | BPC-fKL (sghmc) | | Ours | |
|---|---|---|---|---|---|---|---|---|---|---|---|---|
| | | | Acc(↑) | NLL(↓) | Acc(↑) | NLL(↓) | Acc(↑) | NLL(↓) | Acc(↑) | NLL(↓) | Acc(↑) | NLL(↓) |
| MNIST | 1 | 0.017 | 74.8 ± 1.17 | 1.90 ± 0.01 | 83.59 ± 1.49 | 1.91 ± 0.02 | 90.46 ± 1.5 | 1.54 ± 0.03 | 82.98 ± 2.2 | 1.87 ± 0.03 | **93.42 ± 0.09** | **1.53 ± 0.01** |
| | 10 | 0.17 | 95.27 ± 0.17 | 1.53 ± 0.01 | 91.72 ± 0.55 | 1.52 ± 0.01 | 89.8 ± 0.82 | 1.52 ± 0.01 | 92.05 ± 0.42 | **1.51 ± 0.02** | **97.71 ± 0.24** | 1.57 ± 0.02 |
| | 50 | 0.83 | 94.18 ± 0.26 | 1.36 ± 0.02 | 93.72 ± 0.55 | 1.48 ± 0.02 | 95.58 ± 1.63 | 1.37 ± 0.02 | 40.63 ± 1.8 | 1.36 ± 0.02 | **98.91 ± 0.22** | **1.36 ± 0.01** |
| FMNIST | 1 | 0.017 | 70.53 ± 1.09 | 2.47 ± 0.02 | 72.39 ± 0.87 | 2.15 ± 0.01 | **78.24 ± 1.02** | 1.95 ± 0.04 | 72.51 ± 2.53 | 2.30 ± 0.02 | 77.29 ± 0.5 | **1.90 ± 0.03** |
| | 10 | 0.17 | 78.81 ± 0.17 | 1.64 ± 0.01 | 83.69 ± 0.51 | 1.64 ± 0.03 | 82.06 ± 0.44 | **1.53 ± 0.02** | 83.29 ± 0.55 | 1.54 ± 0.03 | **88.40 ± 0.21** | 1.56 ± 0.01 |
| | 50 | 0.83 | 76.97 ± 0.59 | 1.48 ± 0.02 | 74.41 ± 0.48 | 1.52 ± 0.03 | 82.40 ± 0.35 | 1.32 ± 0.02 | 74.82 ± 0.52 | 1.47 ± 0.02 | **89.47 ± 0.06** | **1.30 ± 0.02** |
| SVHN | 1 | 0.014 | 18.34 ± 1.79 | 3.01 ± 0.02 | 33.52 ± 1.15 | 2.89 ± 0.01 | 48.02 ± 5.62 | 2.44 ± 0.03 | 21.48 ± 6.58 | 2.57 ± 0.02 | **66.74 ± 0.09** | **2.38 ± 0.04** |
| | 10 | 0.14 | 60.68 ± 5.07 | 2.00 ± 0.01 | 74.75 ± 1.27 | 1.95 ± 0.02 | 65.64 ± 2.92 | 2.13 ± 0.01 | 75.49 ± 0.84 | 1.84 ± 0.01 | **82.32 ± 0.56** | **1.81 ± 0.01** |
| | 50 | 0.7 | 78.27 ± 0.62 | 1.89 ± 0.01 | 79.49 ± 0.54 | 1.90 ± 0.01 | 79.6 ± 0.53 | 1.86 ± 0.01 | 77.08 ± 1.8 | **1.72 ± 0.01** | **88.41 ± 0.12** | 1.88 ± 0.02 |
| Cifar10 | 1 | 0.02 | 21.62 ± 0.83 | 2.57 ± 0.01 | 29.34 ± 1.21 | 2.14 ± 0.03 | 35.57 ± 0.95 | 1.97 ± 0.03 | 29.3 ± 1.1 | 2.10 ± 0.03 | **46.87 ± 0.2** | **1.87 ± 0.02** |
| | 10 | 0.2 | 37.89 ± 1.54 | 2.13 ± 0.02 | 48.9 ± 1.72 | 1.73 ± 0.02 | 43.07 ± 1.06 | 1.89 ± 0.02 | 49.85 ± 1.37 | 1.73 ± 0.01 | **56.39 ± 0.7** | **1.72 ± 0.03** |
| | 50 | 1 | 37.54 ± 1.32 | 1.93 ± 0.03 | 46.17 ± 0.67 | 1.62 ± 0.02 | 50.92 ± 1.49 | 1.70 ± 0.03 | 42.30 ± 2.87 | **1.54 ± 0.01** | **71.93 ± 0.17** | 1.57 ± 0.03 |
| Cifar100 | 1 | 0.2 | 3.56 ± 0.04 | 4.69 ± 0.02 | 12.19 ± 0.22 | 4.20 ± 0.01 | 7.57 ± 0.54 | 4.25 ± 0.04 | 12.07 ± 0.16 | 4.27 ± 0.02 | **23.97 ± 0.11** | **4.01 ± 0.02** |
| | 10 | 2 | - | | - | | - | | - | | **28.42 ± 0.24** | **3.14 ± 0.02** |
| T-ImageNet | 1 | 0.2 | - | | - | | - | | - | | **8.39 ± 0.07** | **4.72 ± 0.01** |
| | 10 | 2 | - | | - | | - | | - | | **17.82 ± 0.39** | **3.64 ± 0.05** |

2009), SVHN (Sermanet et al., 2012), MNIST (LeCun et al., 1998), FashionMNIST (Xiao et al., 2017), CIFAR100 (Krizhevsky & Hinton, 2009) and Tiny Imagenet (T-Imagenet) (Le & Yang, 2015). All the experiments perform multi-class classification tasks with ipc=1, 10, and 50 which is in line with previous baselines.We employ Langevin dynamics (Neal et al., 2011b; Teh et al., 2003) during training as well as inference and report accuracy (Acc) and negative log-likelihood (NLL) with respect to the ground truth labels. For our primary experiments, we use a CNN architecture (ConvNet) exactly as described in the previous works (Kim et al., 2022b; Manousakas et al., 2020; Cazenavette et al., 2023) (cf. Appendix for details) for a fair comparison.

Further, we assess the robustness of the BPC methods on out-of-distribution dataset in Section 4.4. We also examine the cross-architecture performance of the proposed method in Section 4.5. Lastly, since bayesian methods are often sensitive to the number of parameters being sampled from the posterior, we observe the effect of number of parameters on the proposed method and compare it with previous BPC baselines in Section 4.6. We refer the reader to Appendix for details regarding these experiments.

## 4.2 BASELINES AND COMPARISONS

We consider the state-of-the-art BPC methods using reverse-KL (BPC-rKL), forward-KL (BPC-fKL), and Wasserstein distance (BPC-W) (Kim et al., 2022a; Manousakas et al., 2020) for comparison. Further comparison with other coreset methods and dataset condensation is provided in the Appendix. All the baselines are implemented using the official codebase provided by respective methods if available, otherwise, we directly take the reported numbers. In cases, neither the codebase nor the numbers are reported, we exclude them from our tables.

## 4.3 RESULTS AND COMPARISON

Table 1 presents the results of the comparative analysis between our approach and other BPC baselines. We observe that the proposed method significantly outperforms all the BPC baselines by large margins. For instance, we observe an improvement of $11.3\%$, $6.54\%$, and $21.01\%$ in accuracy for CIFAR10 with ipc values of 1, 10, and 50, respectively. Additionally, there is a decrease of 0.1 and 0.01 points in negative log-likelihood for ipc values of 1 and 10, respectively, in comparison to the best-performing BPC baseline. Similarly, on SVHN, we notice an improvement in accuracy and negative log-likelihood. Specifically, we observe gains of $18.72\%$, $6.83\%$ in accuracy and reduction of 0.06, 0.03 point in negative log-likelihood for ipc 1, 10 respectively, compared to the BPC counterparts. A similar trend can be seen for MNIST and FMNIST as well. We attribute this boost in performance to the flexible formulation of the proposed method.

We present the qualitative visualizations for MNIST, FMNIST, SVHN, and CIFAR10 datasets with 1 image per class and 10 image per class in Fig. 1. It can be seen that the constructed pseudo-coreset is identifiable but inherits some artifacts due to the constraints on the dataset size. As the number of images per class increases, the model can induce more variations across all the classes and thus

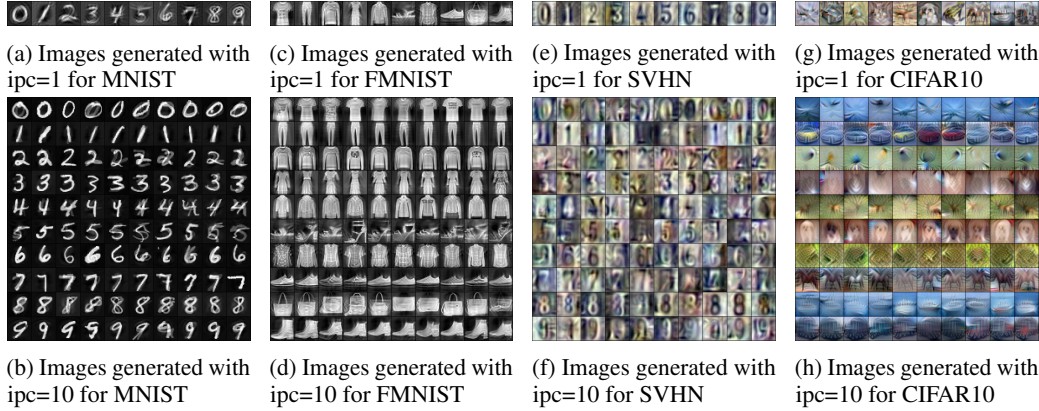

(a) Images generated with ipc=1 for MNIST

(c) Images generated with ipc=1 for FMNIST

(e) Images generated with ipc=1 for SVHN

(g) Images generated with ipc=1 for CIFAR10

(b) Images generated with ipc=10 for MNIST

(d) Images generated with ipc=10 for FMNIST

(f) Images generated with ipc=10 for SVHN

(h) Images generated with ipc=10 for CIFAR10

Figure 1: Visualizations of pseudo-coreset generated from our method with one image per class (top) and ten images per class (bottom) for MNIST, FMNIST, SVHN and CIFAR10. It can be seen that the class labels are identifiable to a large extent.

Table 2: Comparison of the proposed method with BPC baselines for the performance on out-of-distribution data. The classifier model is trained on pseudo-coresets generated using CIFAR10. However, the model is evaluated on CIFAR10-C dataset with different types of corruption.

| Corruption | BPC-rKL (sghmc) | | BPC-W (sghmc) | | BPC-fKL (hmc) | | BPC-fKL (sghmc) | | Ours | |
|---|---|---|---|---|---|---|---|---|---|---|
| | Acc(↑) | NLL(↓) | Acc(↑) | NLL(↓) | Acc(↑) | NLL(↓) | Acc(↑) | NLL(↓) | Acc(↑) | NLL(↓) |
| Gaussian Blur | 31.02 ± 2.65 | 2.132 ± 0.77 | 35.66 ± 1.21 | 2.04 ± 0.12 | 34.76 ± 1.86 | 1.899 ± 0.0436 | 39.73 ± 2.72 | 1.94 ± 0.051 | **41.36 ± 0.72** | **1.73 ± 0.83** |
| Gaussian Noise | 25.49 ± 1.89 | 2.28 ± 0.08 | 33.21 ± 0.89 | 2.11 ± 0.03 | 36.7 ± 1.0 | 1.86 ± 0.02 | 35.71 ± 2.29 | 2.00 ± 0.05 | **38.01 ± 1.26** | **1.82 ± 0.11** |
| JPEG Compression | 30.40 ± 0.9 | 2.13 ± 0.02 | 26.33 ± 1.34 | 2.26 ± 0.04 | 36.2 ± 1.92 | 1.85 ± 0.03 | 37.26 ± 2.87 | 1.95 ± 0.06 | **37.33 ± 0.19** | **1.71 ± 0.03** |
| Snow | 26.85 ± 1.71 | 2.2 ± 0.07 | 37.5 ± 3.5 | 1.93 ± 0.08 | 33.99 ± 1.91 | **1.91 ± 0.03** | 35.68 ± 2.71 | 2.0 ± 0.07 | **37.84 ± 0.64** | 1.91 ± 0.05 |
| Impulsive Noise | 28.39 ± 1.48 | 2.15 ± 0.06 | 36.71 ± 1.93 | 1.96 ± 0.05 | 33.81 ± 1.58 | 1.94 ± 0.02 | **38.26 ± 2.34** | 1.92 ± 0.05 | 37.98 ± 2.15 | **1.89 ± 0.07** |
| Zoom Blur | 31.74 ± 1.24 | 2.09 ± 0.04 | 36.22 ± 2.08 | 1.99 ± 0.05 | 31.3 ± 3.64 | 1.98 ± 0.08 | 35.05 ± 2.90 | 2.04 ± 0.07 | **38.30 ± 0.77** | **1.93 ± 0.13** |
| Pixelate | 28.98 ± 2.26 | 2.19 ± 0.07 | 27.98 ± 1.77 | 2.20 ± 0.05 | 35.59 ± 1.94 | **1.88 ± 0.03** | **39.14 ± 3.15** | 1.93 ± 0.06 | 38.97 ± 1.51 | 1.92 ± 0.07 |
| Speckle Noise | 29.88 ± 0.59 | 2.09 ± 0.02 | 33.33 ± 2.18 | 2.05 ± 0.05 | 34.37 ± 2.02 | 1.90 ± 3.57 | 40.54 ± 1.93 | **1.89 ± 0.04** | **42.66 ± 0.83** | 1.95 ± 0.03 |
| Defocus Blur | 27.57 ± 1.31 | 2.20 ± 0.05 | 33.80 ± 4.21 | 2.09 ± 0.11 | 33.6 ± 2.93 | 1.93 ± 0.06 | 36.72 ± 3.68 | 1.99 ± 0.08 | **37.15 ± 1.03** | **1.87 ± 0.04** |
| Motion Blur | 17.38 ± 2.51 | 2.73 ± 0.14 | 35.22 ± 3.35 | 2.01 ± 0.08 | 34.33 ± 1.89 | 1.92 ± 0.042 | 35.24 ± 3.30 | 2.01 ± 0.05 | **37.06 ± 0.49** | **1.92 ± 0.04** |

produce a diverse pseudo-coreset. Additional qualitative visualizations for pseudo-coreset generated with 50 images per class on CIFAR100 and T-ImageNet dataset are presented in the Appendix.

## 4.4 RESULTS ON OUT OF DISTRIBUTION (OOD) DATASET

We present the results of the proposed method on out-of-distribution (OOD) dataset in Table 2. We use CIFAR10-C (Hendrycks & Dietterich, 2019) dataset for this experiment. In particular, we sample the parameters from the pseudo-coreset posterior obtained using clean CIFAR10 (ipc=10) and perform inference on the corrupted CIFAR10-C, which consists of CIFAR10 images afflicted with different types of corruption including Gaussian Blur, Gaussian Noise, etc. It is evident from Table 2 that our method demonstrates robustness to various types of corruption and exhibits superior performance compared to other baselines. Notably, for corruptions like Gaussian Blur, our method achieves a 1.63% increase in accuracy and a 0.17-point reduction in negative log-likelihood compared to the best-performing BPC baseline. Likewise, for JPEG Compression, Zoom Blur, and Defocus Blur, our method yields an improvement of 0.07%, 2.08%, and 0.43% in accuracy, along with a reduction of 0.14, 0.05, and 0.06 points in negative log-likelihood, respectively. The robustness of the proposed method to different forms of corruption highlights its ability to provide a better approximation of underlying posterior distribution when compared to other baselines.

## 4.5 RESULTS ON CROSS-ARCHITECTURE EXPERIMENTS

Here, we present the cross-architecture results pertaining to various BPC methods. In these experiments, we construct the pseudo-coreset using the said ConvNet model, while during inference, we use different architectures such as ResNet (He et al., 2016b), VGG-Net (Simonyan & Zisserman, 2014) and AlexNet (Krizhevsky et al., 2017a) for evaluation. We perform these experiments for CIFAR10 (ipc = 10). The results of the cross-architecture experiments are presented in Table 3. It can be seen that previous BPC methods fail to generalize across different network architectures, whereas our method demonstrates the ability to adapt to various architectures. For instance, the performance

Table 3: Cross-architecture generalization analysis of BPC methods.

|  | ConvNet | ResNet | VGG | AlexNet |
|---|---|---|---|---|
| **Ours** | **56.39 ± 0.7** | **41.65 ± 1.03** | **47.51 ± 0.89** | **30.58 ± 1.43** |
| **BPC-fKL(hmc)** | 44.34 ± 1.11 | 10.15 ± 0.21 | 10.43 ± 0.33 | 10.0 ± 0.0 |
| **BPC-rKL(sghmc)** | 34.48 ± 0.48 | 10.06 ± 0.08 | 10.26 ± 0.35 | 10.0 ± 0.0 |

Table 4: Performance comparison of the proposed method and other BPC baselines for different parameterized architectures.

| Methods | CN-D3W128 320,010 | CN-D3W256 1,229,834 | CN-D5W128 596,490 | AlexNet 1,872,202 | VGG11 9,231,114 | ResNet 11,173,962 |
|---|---|---|---|---|---|---|
| **Ours** | **56.39 ± 0.70** | **55.93 ± 1.30** | **56.01 ± 0.69** | **52.88 ± 1.39** | **49.26 ± 2.33** | **48.67 ± 0.52** |
| **BPC-rKL (sghmc)** | 37.89 ± 1.54 | 35.82 ± 1.88 | 35.92 ± 1.88 | 32.60 ± 1.45 | 27.66 ± 0.73 | 24.98 ± 1.53 |
| **BPC-W (sghmc)** | 48.90 ± 1.72 | 43.71 ± 1.42 | 46.01 ± 0.92 | 39.01 ± 0.51 | 35.11 ± 1.82 | 32.84 ± 1.38 |
| **BPC-fKL (hmc)** | 49.85 ± 1.37 | 45.87 ± 0.78 | 47.92 ± 1.27 | 41.22 ± 1.62 | 37.05 ± 1.24 | 35.10 ± 2.03 |

of BPC-fKL and BPC-rKL drop by $34.19\%$ and $24.42\%$ respectively on ResNet, resulting in random predictions with an accuracy of almost $10\%$, whereas our method observes a drop of only $14.74\%$ while giving an accuracy of $41.65\%$.

### 4.6 EFFECT OF DIFFERENT NUMBER OF PARAMETERS

Lastly, we analyze the performance of BPC methods across differently parameterized networks. Specifically, we generate pseudo-coresets for CIFAR10 (ipc=10) by employing ConvNets with different parameter configurations. These configurations encompass ConvNets with different depth and width. We also conduct a comparative analysis with other deep learning architectures, including AlexNet (Krizhevsky et al., 2017a), VGG11 (Simonyan & Zisserman, 2014), and ResNet (He et al., 2016b). Bayesian inference techniques generally encounter scalability issues when dealing with large parametric networks (Jospin et al., 2022). This experiment is conducted to ascertain the impact of both large and small architectures on the performance of pseudo-coresets.

The results for different parameterized architectures are presented in Table 4. Here, CN-DxWy denotes a ConvNet architecture with a depth of 'x' and width of 'y'. It is evident from the results that the performance of all BPC methods declines as the number of parameters in the architectures increases. However, our model exhibits relatively better performance in comparison to other methods. Specifically, while our method demonstrates a $7.72\%$ decrease in performance for the ResNet architecture, other BPC baselines such as BPC-fKL, BPC-W, and BPC-rKL experience declines of approximately $14.75\%$, $16.06\%$, and $12.91\%$, respectively. This observation underscores the greater tolerance of our method to large parametric models when compared to other baselines. We again highlight that the performance gain achieved by our proposed method can be attributed to the current formulation, which can generate better approximation to the true posterior.

## 5 CONCLUSION

In this work, we propose a novel approach to generate pseudo-coreset using contrastive divergence. Our approach addresses the need to approximate the posterior of pseudo-coreset and uses a finite number of steps in MCMC methods to sample the parameters from the underlying posterior distribution. Subsequently, these parameters are used to construct pseudo-coreset via contrastive divergence. The empirical evidence presented in our study illustrates that our proposed method surpasses previous BPC baselines by substantial margins across multiple datasets.

***Limitations and Future Work***: While our approach effectively removes variational assumptions associated with the pseudo-coreset posterior and utilizes MCMC methods for parameter sampling, our study still relies on certain assumptions about the posterior of the original dataset. Since there remains a significant performance gap between the pseudo-coreset and the original dataset, a potential avenue for future research could be to relax these assumptions to enhance the performance of BPC methods.

***Broader Impact***: BPC methods have positive applications in democratization and privacy-related concerns by reducing the dependence on the original dataset. We don't believe that our method has any associated negative societal impact.

***Reproducibility Statement***: To ensure that the proposed work is reproducible, we have included an Algorithm (cf. Algorithm 1) in the Appendix. We have included the training details and hyperparameters in Section A.1 in the Appendix. We have clearly defined our loss function in Eq. 10. The code for the proposed method can be found at https://anonymous.4open.science/r/BPC-CD-E762.

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

# A   APPENDIX

## A.1   TRAINING DETAILS AND HYPER PARAMETERS

In this section, we provide the implementation details of the proposed method. Our implementation can be found https://anonymous.4open.science/r/BPC-CD-E762. During the training process, we randomly initialize the synthetic dataset using samples from the original training set. The overall cardinality of these synthetic sets is determined by the number of images considered for every class (ipc). For our experiments, we have considered ipc values of 1, 10 and 50. Furthermore, similar to previous works Kim et al. (2022a); Cazenavette et al. (2022), we have used Differentiable Siamese Augmentation (DSA) (Zhao & Bilen, 2021) strategies to enhance the performance of our model. DSA strategies include random crop, random flip, random brightness, random scale, and rotation. At any instant, we apply one of these augmentations to train our network. These augmentation techniques ensure that the model does not overfit on the given synthetic set and generates optimal parameters. DSA is applied to the synthetic set while running langevin dynamics and calculating the contrastive-divergence-based loss function.

We have also conducted additional experiments to evaluate the effectiveness of our method and other BPC baselines in the absence of DSA. The findings of the experiment are reported in Table 5. The results clearly indicate that the DSA has a positive impact on the performance of all BPC methods, which align with the observation made by Cui et al. (2022). Nevertheless, even without the DSA-based augmentation strategy, our method outperforms other BPC baselines for 1 ipc.

As for the network used to calculate the energy, we take inspiration from previous works (Kim et al., 2022a; Wang et al., 2018; Lee et al., 2022; Cazenavette et al., 2022) and use a ConvNet architecture (Gidaris & Komodakis, 2018). This architecture consists of multiple blocks of convolutional layer with filter dimension of $3 \times 3$ and channel size of $128$. The network uses instance normalization, maxpool layer with stride 2, and RELU activation. In our experiments, we have used an architecture with three such blocks of convolution layers.

Next, we create a buffer of trajectories to sample parameters from the posterior of the original dataset. For this, we generate 100 different trajectories, each with 50 epochs trained using SGD optimizer with a batch size of 256 on the original training set. These parameters are used to obtain the gaussian variational approximation to estimate the loss function. Further, we use diagonal covariance matrix with diagonal entry of $0.001$ for the re-parameterization trick used in gaussian approximation.

The pseudocode of the implementation is presented in Algorithm 1. The hyperparameters used are as follows: $P = 2000$, $\lambda = 0.01$, $n = 50$, $L = 100$, $\Sigma_{\mathbf{x}}^{1/2} = 0.001I$. These hyper-parameters are fixed across all the datasets. Further, we observe that there $\gamma$ that can be fine-tuned for marginal improvements in performance. Specifically, $\gamma$ is varied between $\{1, 10, 100, 1000\}$. Note that, we use same set of parameter to sample parameters during inference. All the experiments are conducted on a single NVIDIA RTX A6000 GPUs with 48GB memory.

Table 5: Comparison of the proposed method against other BPC baselines without using DSA on CIFAR10 dataset.

| BPC-rkl (sghmc) | | BPC-w (sghmc) | | BPC-fkl (hmc) | | BPC-fkl (sghmc) | | Ours | |
|---|---|---|---|---|---|---|---|---|---|
| ipc =1 | ipc = 10 | ipc =1 | ipc = 10 | ipc =1 | ipc = 10 | ipc =1 | ipc = 10 | ipc =1 | ipc = 10 |
| $19.70 \pm 1.06$ | $36.41 \pm 0.75$ | $27.66 \pm 0.8$ | $39.61 \pm 1.12$ | $32.61 \pm 1.50$ | $38.12 \pm 1.19$ | $28.25 \pm 0.92$ | $\mathbf{41.85 \pm 1.47}$ | $\mathbf{34.94 \pm 0.72}$ | $41.02 \pm 0.66$ |

---

**Algorithm 1** Proposed Algorithm

---

**Input :** Set of SGD trajectories obtained from original dataset ($\tau$), Number of langevin steps ($L$) needed to sample parameter from $\pi_{\tilde{\mathbf{x}}}$, Langevin step size ($\lambda$), Step size to modify pseudo-coreset ($\gamma$), Number of epochs (P)

1: Initialize pseudo-coreset ($\tilde{x}$) using samples from original dataset $x$.
2: **for** step in [1... P] : **do**
3:      Sample $\tau_i \sim \tau$
4:      Sample $\theta_k^+ \sim \tau_i$ where $\theta_k^+$ are parameters associated with $k^{th}$ epoch for $i^{th}$ trajectory.
5:      Let $\theta^+ = \theta_k^+ + \Sigma_{\mathbf{x}}^{1/2}\varepsilon_{\mathbf{x}}, \varepsilon \sim \mathcal{N}(0, I)$
6:      Let $\theta_0^- = \theta^+$
7:      **for** t in [0 .... L] : **do**
8:          Calculate energy associated with $\tilde{\mathbf{x}}$ and parameter $\theta_t^-$ i.e. $E(\theta_t^-, \tilde{\mathbf{x}})$
9:          $\theta_{t+1}^- = \theta_t^- - \lambda(\nabla_\theta E(\theta_t^-, \tilde{\mathbf{x}})) + \eta, \eta \sim \mathcal{N}(0, I)$
10:      Let $\theta^- = \theta_L^-$
11:      Calculate $\mathcal{L} = E((\theta^+, \tilde{\mathbf{x}})) - E((\theta^-, \tilde{\mathbf{x}}))$
12:      $\tilde{\mathbf{x}} \leftarrow \tilde{\mathbf{x}} - \gamma\nabla_{\tilde{\mathbf{x}}}\mathcal{L}$

---

## A.2 EXPERIMENTAL SETUP

### A.2.1 BASELINE SETUP

We primarily present the results for different BPC frameworks. The experiment of Table 1 in the main manuscript uses the original hyperparameters mentioned in the respective papers. In cases where hyperparameters were not explicitly specified, we employed the default hyperparameters of CIFAR10. We have presented the results for BPC methods with only 1 and 10 ipc for the CIFAR100 and T-ImageNet datasets. We could not report the result for other scenarios due to the computational limitations. These methods demand a significant amount of GPU memory, which we currently lack, making it impractical to compute the desired results.

### A.2.2 GPU AND TIME CONSUMPTION

We assess the computational efficiency of our method relative to other baselines by comparing the GPU memory consumption and the training time required to generate the pseudo-coresets for a single iteration. The findings of our results are presented in Fig. 2, where the iteration time is calculated by taking the average of the total time for 100 different iterations.

As illustrated in Fig. 2, our method requires relatively less time compared to other BPC methods for low ipc values and outperforms BPC-W for higher ipc values. Additionally, in our examination of GPU memory usage, we observe that BPC-W shows linear scaling in GPU memory consumption as the number of images per class increases. In contrast, our method maintains consistent memory usage across all ipc values. In our experiment, we found that our method utilizes only 37GB of memory, even for higher images per class. It's worth noting that other BPC baselines such as BPC-fKL and BPC-rKL are more memory-efficient than our method and deliver consistent performance across

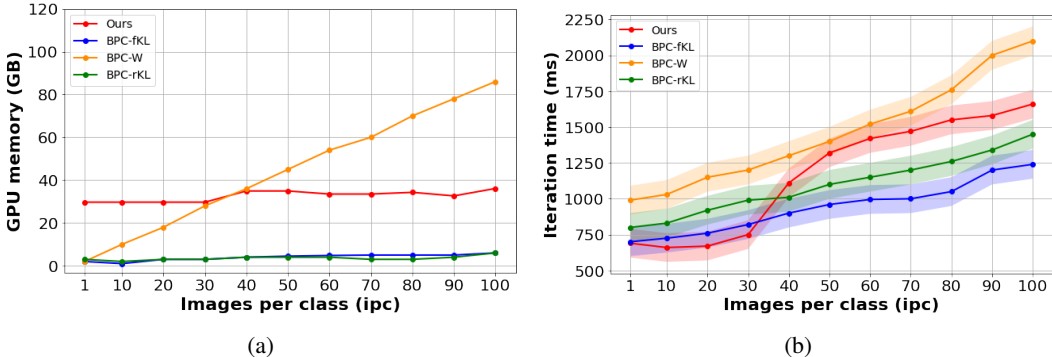

Figure 2: Computing GPU memory costs along with training time for different image per class.

all the ipc values. We attribute this observation to the fact that BPC-fkl and BPC-rkl avoid MCMC sampling during training by making use of relevant approximations. Whereas, our method makes use of gradient-based MCMC sampling (langevin dynamics) for estimation of the objective function. For this reason, the GPU consumption of the proposed method is relatively higher than that of BPC-fkl and BPC-rkl.

### A.3 COMPARISON WITH CORESET METHODS

We have conducted a comparative analysis of our method with other coreset techniques such as Herding (Chen et al., 2012), K-Center (Sener & Savarese, 2017), and Forgetting (Toneva et al., 2018). The outcomes of our experiments are listed in Table 6. The result clearly shows that our method outperforms other coreset techniques on all the dataset.

Table 6: Comparison of the proposed method with coreset baselines. The results are noted in form of (mean $\pm$ std. dev) where we have obtained test accuracy over five independent runs on the pseudo-coreset. The best performer across all methods is denoted in bold ($x \pm s$). For ease of comparison, we color the second best performer with blue color.

|  | ipc | Ratio(%) | Herding | K-Center | Forgetting | Ours |
|---|---|---|---|---|---|---|
| **MNIST** | 1 | 0.017 | $89.2 \pm 1.6$ | $89.3 \pm 1.5$ | $35.5 \pm 5.6$ | $93.42 \pm 0.09$ |
|  | 10 | 0.17 | $93.7 \pm 0.3$ | $84.4 \pm 1.7$ | $68.1 \pm 3.3$ | $97.71 \pm 0.24$ |
|  | 50 | 0.83 | $94.8 \pm 0.2$ | $97.4 \pm 0.3$ | $88.2 \pm 1.2$ | $98.91 \pm 0.22$ |
| **FMNIST** | 1 | 0.017 | $67.0 \pm 1.9$ | $66.9 \pm 1.8$ | $42.0 \pm 5.5$ | $77.29 \pm 0.5$ |
|  | 10 | 0.17 | $71.1 \pm 0.7$ | $54.7 \pm 1.5$ | $53.9 \pm 2.0$ | $88.40 \pm 0.21$ |
|  | 50 | 0.83 | $71.9 \pm 0.8$ | $68.3 \pm 0.8$ | $55.0 \pm 1.1$ | $89.47 \pm 0.06$ |
| **SVHN** | 1 | 0.014 | $20.9 \pm 1.3$ | $21.0 \pm 1.5$ | $12.1 \pm 1.7$ | $66.74 \pm 0.09$ |
|  | 10 | 0.14 | $50.5 \pm 3.3$ | $14.0 \pm 1.3$ | $16.8 \pm 1.2$ | $82.32 \pm 0.56$ |
|  | 50 | 0.7 | $72.6 \pm 0.8$ | $20.1 \pm 1.4$ | $27.2 \pm 1.5$ | $88.41 \pm 0.12$ |
| **Cifar10** | 1 | 0.02 | $21.5 \pm 1.2$ | $21.5 \pm 1.3$ | $13.5 \pm 1.2$ | $46.87 \pm 0.2$ |
|  | 10 | 0.2 | $31.6 \pm 0.7$ | $14.7 \pm 0.9$ | $23.3 \pm 1.0$ | $56.39 \pm 0.7$ |
|  | 50 | 1 | $23.3 \pm 1.0$ | $27.0 \pm 1.4$ | $23.3 \pm 1.1$ | $71.93 \pm 0.17$ |
| **Cifar100** | 1 | 0.2 | $8.4 \pm 0.3$ | $8.3 \pm 0.3$ | $4.5 \pm 0.2$ | $23.97 \pm 0.11$ |
|  | 10 | 2 | $17.3 \pm 0.3$ | $7.1 \pm 0.2$ | $15.1 \pm 0.3$ | $28.42 \pm 0.24$ |
| **T-ImageNet** | 1 | 0.2 | $2.8 \pm 0.2$ | $3.03 \pm 0.0$ | $1.6 \pm 0.1$ | $8.39 \pm 0.07$ |
|  | 10 | 2 | $6.3 \pm 0.2$ | $11.38 \pm 0.0$ | $5.1 \pm 0.2$ | $17.82 \pm 0.39$ |

### A.4 COMPARISON WITH DATASET CONDENSATION TECHNIQUES

We have also compared our method with other data condensation (DC) techniques like Distillation (DD) (Wang et al., 2018), Flexible Dataset Distillation (LD) (Bohdal et al., 2020), Gradient Matching (DC) (Zhao et al., 2021), Differentiable Siamese Augmentation (DSA) (Zhao & Bilen, 2021), Distribution Matching (DM) (Zhao & Bilen, 2023), Neural Ridge Regression (KIP) (Nguyen et al., 2021), Condensed data to align features (CAFE) (Wang et al., 2022) and Matching Training Trajectories (MTT) (Cazenavette et al., 2022). The results of the experiment are shown in Table 7. We find that

Table 7: Comparison of the proposed method with dataset-condensation baselines. The results are noted in form of (mean ± std. dev) where we have obtained test accuracy over five independent runs on the pseudo-coreset. The best performer across all methods is denoted in bold ($x \pm s$). For ease of comparison, we color the second best performer with blue color.

| | Img/cls | Ratio% | DD | LD | GM | DSA | DM | CAFE | CAFE+DSA | KIP | MTT | Ours |
|---|---|---|---|---|---|---|---|---|---|---|---|---|
| **MNIST** | 1 | 0.017 | - | 60.6±2.86 | 92.01±0.25 | 87.6±0.07 | 88.89±0.57 | 93.1±0.3 | 90.8±0.5 | 85.46±0.04 | 89.85±0.01 | **93.42±0.09** |
| | 10 | 0.17 | 79.71±8.3 | 87.05±0.5 | 97.58±0.1 | 97.39±0.06 | 96.58±0.11 | 97.2±0.2 | 97.5±0.1 | 97.15±0.11 | 97.7±0.02 | **97.71±0.24** |
| | 50 | 0.83 | - | 93.3±0.3 | 98.81±0.03 | 98.97±0.04 | 98.22±0.05 | 98.6±0.2 | 98.9±0.2 | 98.36±0.08 | 98.6±0.006 | **98.91±0.22** |
| **FMNIST** | 1 | 0.017 | - | - | 70.83±0.01 | 70.45±0.57 | 71.92±0.7 | 77.1±0.9 | 73.7±0.7 | - | 77.14±0.007 | **77.29±0.5** |
| | 10 | 0.17 | - | - | 81.93±0.07 | 84.7±0.11 | 83.25±0.09 | 83.0±0.4 | 83.0±0.3 | - | **88.768±0.00158** | 88.40±0.21 |
| | 50 | 0.83 | - | - | 83.26±0.17 | 88.55±0.56 | 87.65±0.03 | 84.8±0.4 | 88.2±0.3 | - | 89.332±0.151 | **89.47±0.06** |
| **SVHN** | 1 | 0.014 | - | - | 30.49±0.57 | 31.18±0.43 | 19.25±1.39 | 42.6±3.3 | 42.9±3.0 | - | 57.55±0.02 | **66.74±0.09** |
| | 10 | 0.14 | - | - | 75.1±0.4 | 78.39±0.3 | 71.42±1.01 | 75.9±0.6 | 77.9±0.6 | - | 72.56±0.005 | **82.32±0.56** |
| | 50 | 0.7 | - | - | 81.7±0.14 | 82.5±0.34 | 82.41±0.52 | 81.3±0.3 | 82.3±0.4 | - | 83.731±0.334 | **88.41±0.12** |
| **CIFAR10** | 1 | 0.02 | - | 25.38±0.2 | 28.10±0.56 | 29±0.64 | 26.40±0.42 | 30.3±1.1 | 31.6±0.8 | 40.5±0.4 | 46.08±0.8 | **46.87±0.2** |
| | 10 | 0.2 | 39.14±2.3 | 37.5±0.6 | 44.14±0.6 | 51.85±0.43 | 48.66±0.03 | 46.3±0.6 | 50.9±0.5 | 53.1±0.5 | **64.27±0.8** | 56.39±0.7 |
| | 50 | 1 | - | 41.7±0.5 | 53.73±0.44 | 60.77±0.45 | 62.7±0.07 | 55.5±0.6 | 63.3±0.4 | 58.6±0.4 | 71.26±0.5 | **71.93±0.17** |
| **CIFAR100** | 1 | 0.2 | - | 11.5±0.4 | 12.65±0.32 | 13.88±0.29 | 11.35±0.18 | 12.04±0.0 | 12.9±0.3 | 14.0±0.3 | 23.62±0.63 | **23.97±0.11** |
| | 10 | 2 | - | - | 25.28±0.29 | 32.34±0.4 | 29.38±0.26 | 29.04±0.0 | 27.8±0.3 | 31.5±0.2 | **36.96±0.155** | 28.42±0.24 |
| **T-ImageNet** | 1 | 0.2 | - | - | 5.27±0.0 | 5.67±0.0 | 3.82±0.0 | - | - | - | 8.27±0.0 | **8.39±0.07** |
| | 10 | 2 | - | - | 12.83±0.0 | 16.43±0.0 | 13.51±0.0 | - | - | - | **20.11±0.0** | 17.82±0.39 |

the performance of our method is better than almost all the DC baselines, whereas MTT stands out to be a close second in most of the cases. This shows that our method, although falling under the category of Bayesian pseudo-coreset, achieves a performance that is comparable to that of heuristic DC methods. It is to be noted that the DC methods are not the direct competitors of our method. However, we have shown that our method, although a BPC, surpasses (or comes very close to) the SoTA DC methods such as MTT (Cazenavette et al., 2022).

## A.5   Visualizations for CIFAR100 and Tiny-ImageNet

In this section, we present the visualizations for pseudo-coresets of large datasets like CIFAR100 and Tiny-Imagenet datasets. We present generated synthetic images for both 1 and 10 images per class. We provide the visualization for 1 image per class on both datasets in Fig. 3 and Fig. 4, respectively. Fig. 5a and Fig. 5b include visualization for CIFAR100 datasets with 10 ipc wherein each image is divided based on the number of classes. Similarly, we split the image into 50 classes for the Tiny-ImageNet dataset for 10 ipc in Fig. 6 and Fig. 7.

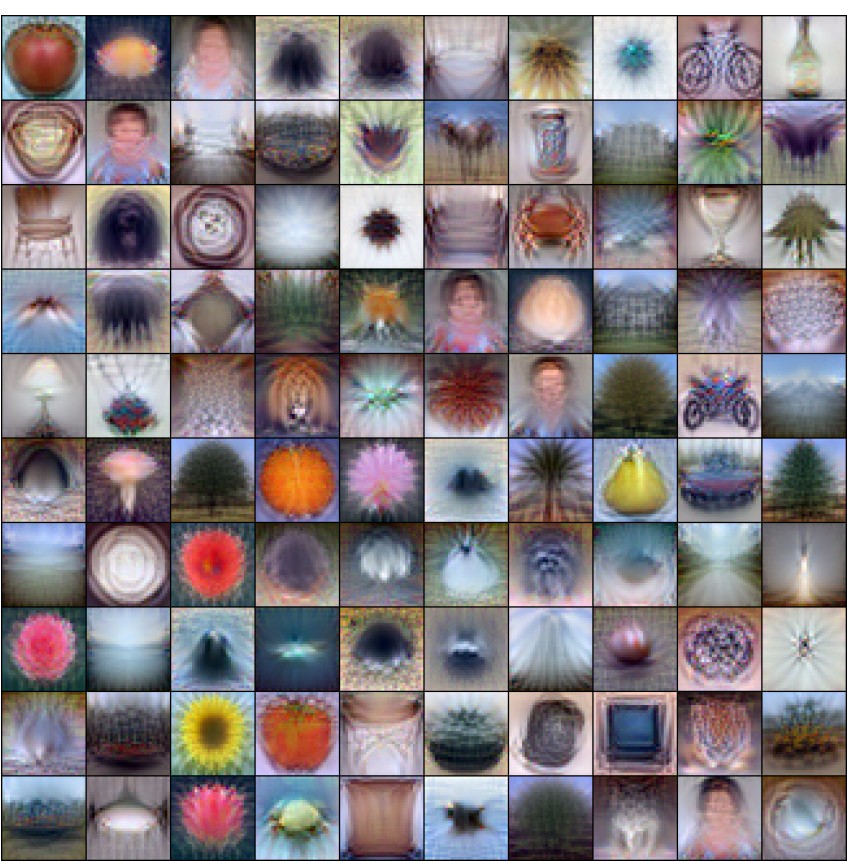

Figure 3: Visualizations of pseudo-coresets for CIFAR100 with 1 Image Per Class.

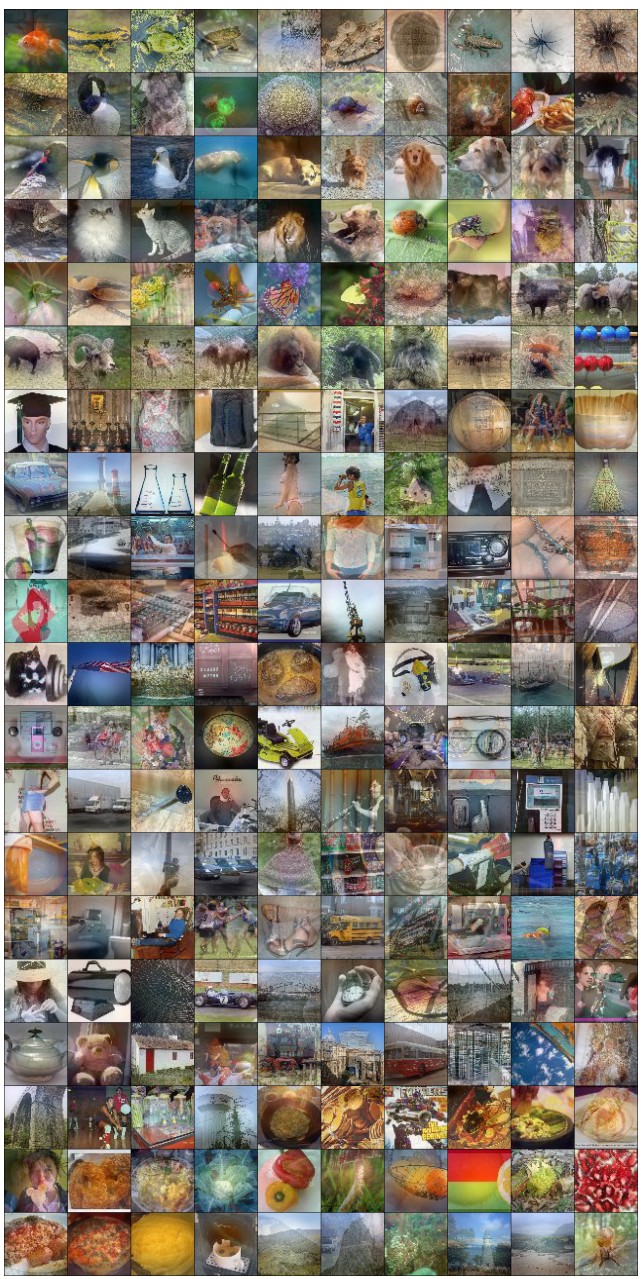

Figure 4: Visualizations of pseudo-coresets for Tiny ImageNet with 1 Image Per Class.

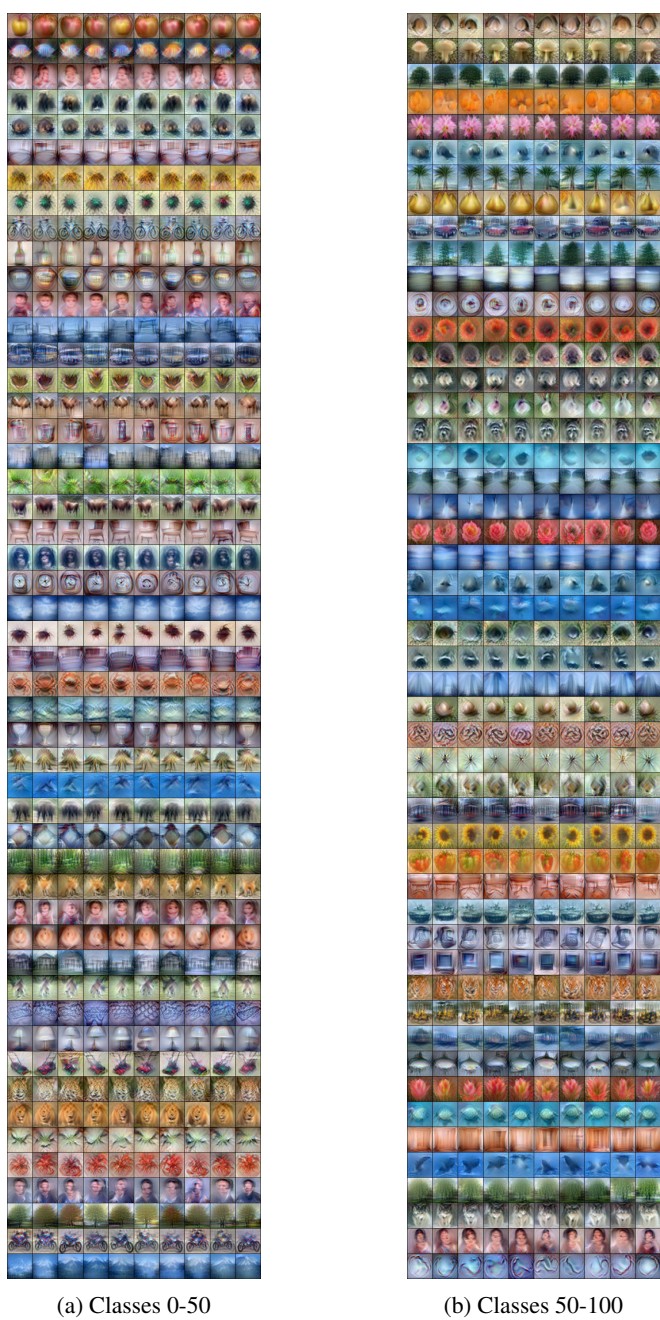

(a) Classes 0-50                                    (b) Classes 50-100

Figure 5: Visualizations for pseudo-coresets for CIFAR100 with 10 Images Per Class

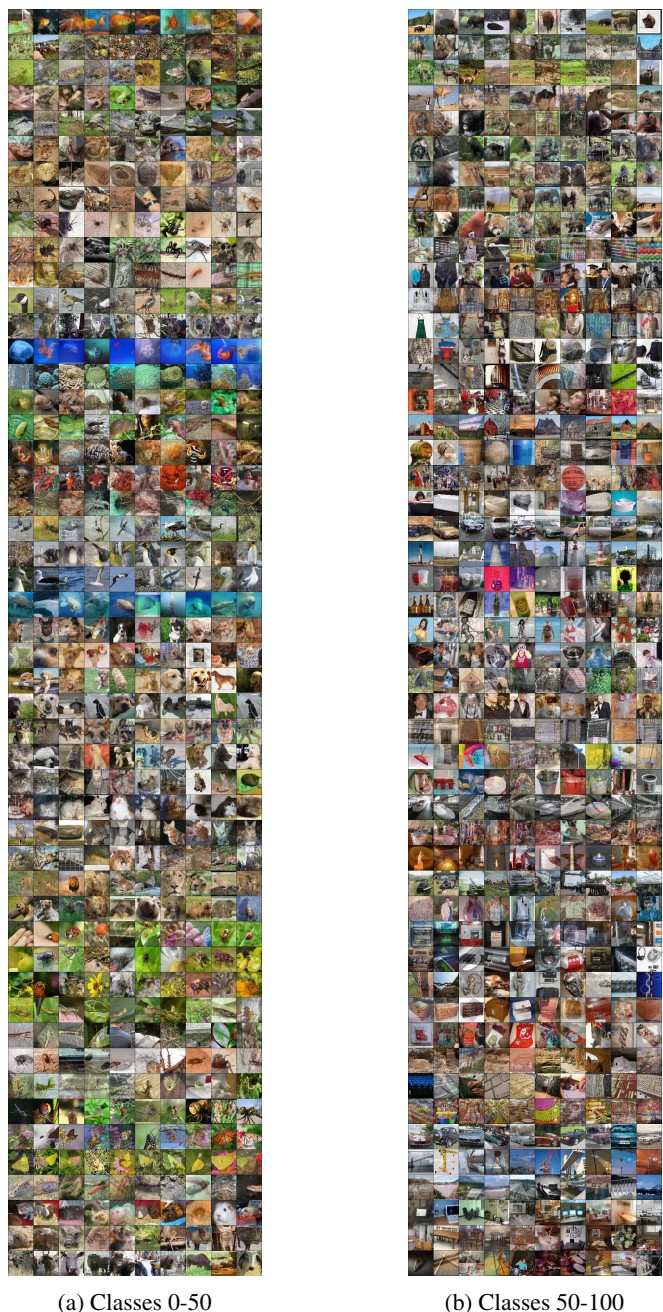

(a) Classes 0-50         (b) Classes 50-100

Figure 6: Visualizations of psuedo-coresets for Tiny ImageNet with 10 Images Per Class

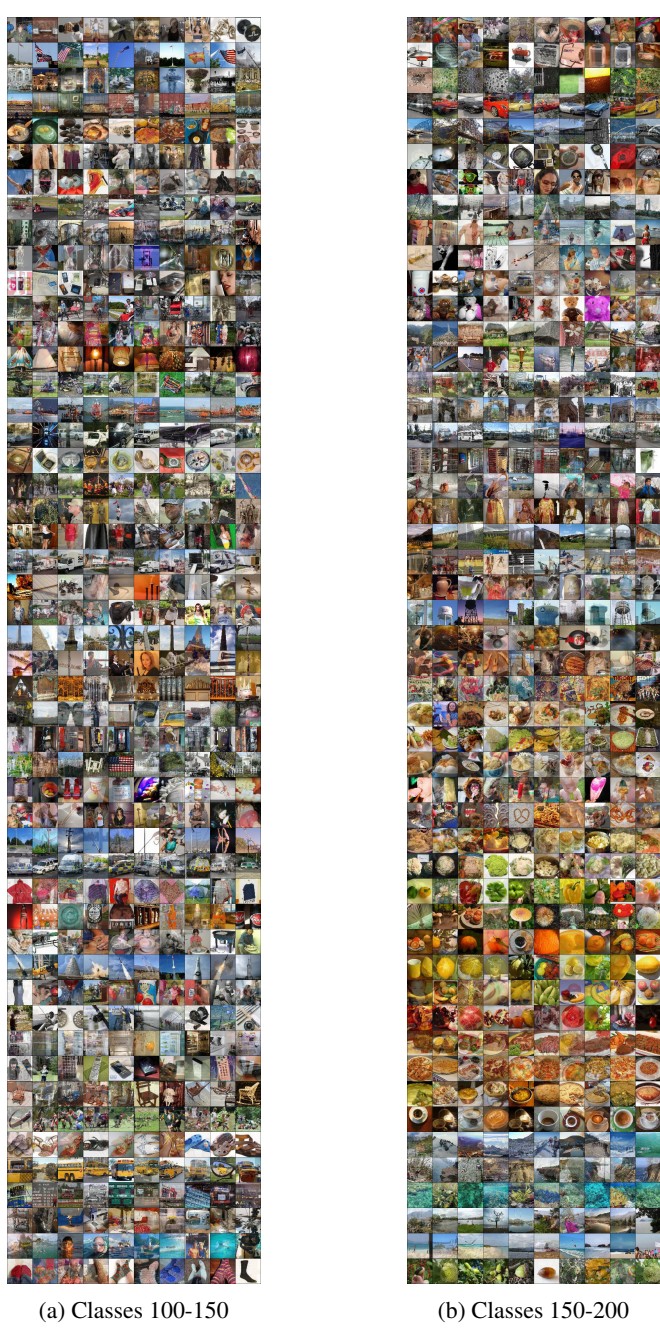

(a) Classes 100-150 (b) Classes 150-200

Figure 7: Visualizations of pseudo-coresets for Tiny ImageNet with 10 Images Per Class

