# OpenReview forum: "Bayesian Pseudo-Coresets via Contrastive Divergence"
_ICLR.cc/2024/Conference — ICLR 2024 Conference Withdrawn Submission_

### Official Review · Reviewer_nxjv · 2023-10-29

**Soundness:** 3 good
**Presentation:** 3 good
**Contribution:** 1 poor
**Rating:** 1
**Confidence:** 5

**Summary:**

The paper discusses the topic of Bayesian pseudo-coresets: a type of distillation for probabilistic models, where a synthetic pseudo-dataset is selected such that the posterior conditioned on them is close to that conditioned on the full dataset. The original optimization for Bayesian pseudo-coreset attempts to minimize a discrepancy measure between the pseudo-posterior $\pi_{\tilde{x}}$ and real posterior $\pi_{x}$, but each gradient step w.r.t. $\tilde{x}$ involves inefficient re-sampling of $\pi_{\tilde{x}}$. The paper proposes to replace the canonical KL divergence measure with the contrastive divergence (CD), such that one could approximate $\pi_{\tilde{x}}$ in a biased fashion via a warm-start simulation of a short MCMC chain. The paper presents sensible empirical results showing that the pseudo-corsets selected via CD minimization yields better results in the distilled dataset.

**Strengths:**

- Soundness: The problem of selecting Bayesian pseudo-coresets is notably similar to training an energy-based model. The method of using contrastive divergence in replacement of KL divergence is well-established and justified.
- Experiments: The paper presents better results compared to other pseudo-coreset selection methods.
- Presentation: The paper is well-written and easy to understand, with a rich literature survey that I find helpful.

**Weaknesses:**

In lieu of the problem of a manual inflation of test accuracy, I'm modifying my view of this paper from the standpoint of integrity.

Here is my original characterization:

I am mainly concerned about the significance of this contribution: the only contribution of this paper is the application of contrastive divergence to this specific problem, which is a natural and incremental next step compared to previous works.

Moreover, there are numerous other divergence measures that could be applied to this problem, many of which share the same advantage as contrastive divergence. I believe that the wholesale introduction and investigation of divergence measures with no / little reliance on long MCMC simulation of $\pi_{\tilde{x}}$ is a significant way to bolster the significance of the paper. I lay out my thoughts in the "questions" section.

**Questions:**

Barp et al. [1] include divergence measures of such sort. As mentioned by this paper, contrastive divergence has been shown to be an approximation of the score-matching divergence, and score-matching divergence can be further smoothed/kernelized into kernel Stein discrepancy - a discrepancy that measures the divergence between 2 distributions, where 1 of them can be represented by the score function. In my opinion, while the introduction of CD can be seen as trivial, a more comprehensive look is an interesting and significant next step for tackling this question.

1. Barp A, Briol FX, Duncan A, Girolami M, Mackey L. Minimum Stein Discrepancy Estimators. In: Advances in Neural Information Processing Systems.

**Details Of Ethics Concerns:**

As pointed out by Reviewer N1MW, the paper's implementation seems to give their method an unfair advantage.

---

> ### Author Response · Authors · 2023-11-13
> **Response on codebase.**
>
> Thank you for your valuable comments. Please find our responses below:
>
> "In lieu of the problem of a manual inflation of test accuracy, I'm modifying my view of this paper from the standpoint of integrity"
>
> Our sincere apologies for this overlooking mistake. Kindly refer to our general comment on the codebase. The reported results doesn't involve any manual accuracy inflation. The uploaded codebase has the wrong version of our implementation. The correct version is available here - https://anonymous.4open.science/r/BPC-CD-E762. We request the Reviewer to kindly check it for reproducibility and correctness.
>
> 1. Further exploration of Contrastive Divergence
>
> Thanks a lot for mentioning this paper, we were not aware of it. We will look into this work and try to build upon and improve our paper.
>
> 2. Regarding the Issue with the Codebase
>
> - Appologies for this confusion. During our  internal review of the codebase, we noticed that we have uploaded a wrong version of the codebase which contained this bug/error. Moreover, this version is also outdated as we had made several changes to it during our experiments. Please refer to the general comment for reference.
> - We have now updated the codebase with the correct version which can be used to generate the results in the paper (https://anonymous.4open.science/r/BPC-CD-E762). We have additionally provided the config files with all the hyperparameters for the ease of verification.

---

### Official Review · Reviewer_N1MW · 2023-10-31

**Soundness:** 1 poor
**Presentation:** 3 good
**Contribution:** 1 poor
**Rating:** 3
**Confidence:** 4

**Summary:**

This paper proposes a method that distills a synthetic dataset (Bayesian Pseudocoreset) whose NN weight posterior approximates the posterior over neural network weights, conditional on an entire training dataset. The proposed method approximates the max. likelihood (=forward kl between target posterior and pseudocoreset posterior) with the contrastive divergence objective and applies several approximations to make the procedure more computationally tractable.

**Strengths:**

- The authors provide plenty of empirical evidence with their work, with a lot of additional results containing experimentation with different architectures, parameterizations and on OOD data.
- Related work section gives a very good and extensive summary of other works in the field.

**Weaknesses:**

- The contributions of this paper are very limited, especially when compared to [1]. The BPC-fKL objective in [1] is very closely related to contrastive divergence, something which is even explained in section 3.3 of [1]. The authors of this paper claim that their method is more efficient because it samples from the pseudocoreset posterior with a finite number of MCMC steps initialized from a sample of the target posterior. However, when comparing their Algorithm 1 to BPC-fKL as described in Algorithm 1 of [1] (and their code), both methods are equivalent apart from the proposed method using noised SGD as the short horizon MCMC kernel instead of SGD in BPC-fKL and not adding any noise to $\theta^-$.

- I see some issues with the provided code, where the code does not align with what is presented in the paper. Line 11 of Algorithm 1 for example is computed with an additional `meta_loss` term in line 464 and 466 of `distill.py`. Furthermore, no noise is added to the pseudocoreset sampler in the code (line 402 of `distill.py`), in contrast to line 9 of Algorithm 1 in the appendix. Additionally, no noise is added to $\theta^+$ (line 351 of `distill.py`), in contrast to line 5 of Algorithm 1. Consequently, the code actually shows an procedure which is even more similar to BPC-fKL from [1] than described in the paper.

- As a result of the mentioned observations in the code, and how the procedure in the code is almost the same as BPC-fKL, I have some doubts about the provided experimental results. Specifically, the margin with which the proposed algorithm beats BPC-fKL seems rather large to me, considering how similar the procedures are. Furthermore, I want to point out the reported accuracy of BPC-fKL in Table 1 for MNIST (sghmc), which decreases to 40.63% for ipc=50, after being 82.98 for ipc=1. In other experiments, I also see the baselines decrease their accuracy for ipc=50 over ipc=10, something which should not happen.

If these weaknesses cannot be refuted by the authors, I would consider this a clear reject.

[1] Kim, B., Choi, J., Lee, S., Lee, Y., Ha, J. W., & Lee, J. (2022). On divergence measures for bayesian pseudocoresets. Advances in Neural Information Processing Systems, 35, 757-767.

**Questions:**

- What are the specific differences between the proposed method and BPC-fKL (as described in Algorithm 1 of [1])?
- Please comment on some of my comments on the code, mentioned above. For example, what is `meta_loss`?
- Why does the accuracy for some of the baselines decrease from ipc=10 to ipc=50 and does not decrease for your proposed method?
- Why does the proposed method perform better with such a large margin over BPC-fKL while being almost equivalent? Can you explain this?
- How does the performance compare for each experiment to a random subset of the data (an ‘untrained’ initialized coreset)?

---

> ### Author Response · Authors · 2023-11-13
>
> Thanks a lot for your comments. Please find our responses in below points:
>
> 1. Difference between BPC-fkl and the proposed method
>
> Thanks for bringing up this point.
>
> - Differences in Objective:  We would like to point that although Kim et. al. talks about their relation with Contrastive-Divergence in Section 3.3 of their paper, the objective that they derive is obtained by differentiating usual forward-KL divergence. Hence, their objective is actually not a true reflection of contrastive-divergence, instead, they are working with usual MLE-based loss function to obtain the pseudo-coreset.
>
> - Differences in Implementation: Implementation wise, the difference between our approach and BPC-fkl is as follows: Both our method and BPC-fkl uses difference between energy function (log-likelihood) at two different parameters, $\theta^+$ and $\theta^-$. BPC-fkl, starts from an initial parameter $\theta_i$, takes $L_x$ steps along the expert trajectory to obtain $\theta^+ = \theta_{i+L_x}$, similarly, they take $L_u$ gradient descent steps on pseudo-coreset starting from $\theta_i$ to obtain $\theta^-$. Lastly, they use gaussian-variational approximation around $\theta^+$ and $\theta^-$ and take the difference between energy evaluated at samples from these approximations. However, in the proposed method, we start from an initial parameter $\theta_i$, fit a gaussian around this parameter and sample from it to obtain $\theta^+$, then we take $L$ steps of langevin-dynamics to obtain $\theta^-$. Lastly, the difference between energies at these two points are used to update the pseudo-coreset.
>
> 2. Regarding the Issue with the Codebase
>
> - Thank you for noting this, and Appologies for this confusion. During our  internal review of the codebase, we noticed that we have uploaded a wrong version of the codebase which contained this additional loss component. Further, this version is also outdated as we made several changes to it during our experiments. Please see the general comment for reference.
> - In particular, the `meta_loss` component was something that we had experimented during initial phase of the work to see if meta-learning based objectives could be used. However, we discarded it in the course of experimentation, the boolean flag for the same was set to false.
> - Further, we follow some of the recent implementations where langevin dynamics is run without adding noise at each step (see e.g. https://github.com/bpucla/latent-space-EBM-prior/blob/a530d0fbfe995cc0dcff14c27170a7cf77492114/train_svhn.py#L487). We observed that, the results were better when the noise is not added, we will mention this in the final draft. The algorithm in the Appendix will be modified to exactly match our implementation.
> - Hence, this version is not the one that was used to generate the final results of the current paper. We sincerely appologise for overlooking this mistake.
>
> - We have now updated the codebase with the correct version which can be used to generate the results in the paper (https://anonymous.4open.science/r/BPC-CD-E762). We have additionally provided the config files with all the hyperparameters for the ease of verification.

---

> > ### Author Response · Authors · 2023-11-13
> >
> > 3. Reduction in Performance for few Baselines
> >
> > Thanks for noticing this. First, we would like to note that the BPC-fkl paper has only reported numbers for ipc=1,10,20. They have not conducted the experiments for larger ipc such as 50, hence we don't have the correct hyperparamters to generate optimal results. We have tried several hyperparameters (including the ones used for our method) and have reported the best performance for each of the baselines. Although, we don't have any theoretical explanation for this observation, we only have empirical results.
> >
> > 4. Superior Performance of the Proposed Method
> >
> > Thanks for this question. We believe that the proposed method performs better than BPC-fkl because we avoid using the variational approximation for the pseudo-coreset posterior. The major intended message of our paper is that the performance of the BPC gets a boost without using a fixed distributional form for the pseudo-coreset posterior, unlike BPC fkl. This trick gives a significant boost to the performance
> >
> > 5. Results with Random Subsets
> >
> > Thanks for this point. We have performed experiments with random subsets. Our observations can be found in the table below:
> >
> >
> >
> > | MNIST (ipc=1) | MNIST (ipc=10) |  MNIST (ipc=50)   | FMNIST (ipc=1)    |  FMNIST (ipc=10)   |  FMNIST (ipc=50)   |  SVHN (ipc=1)   |  SVHN (ipc=10)   |  SVHN (ipc=50)   | CIFAR10 (ipc=1)    |  CIFAR10 (ipc=10)   | CIFAR10 (ipc=50) |
> > | ------------- | ------------- | --- | --- | --- | --- | --- | --- | --- | --- | --- | ------------- |
> > | 64.9 $\pm$ 3.5          | 95.1 $\pm$ 0.9          |  97.9 $\pm$ 0.2   |  51.4 $\pm$ 3.8   |  73.8 $\pm$ 0.7   |  82.5 $\pm$ 0.7   |  14.6 $\pm$ 1.6   |  35.1 $\pm$ 4.1   |  70.9 $\pm$ 0.9   |  14.4 $\pm$ 2.0   |  26.0 $\pm$ 1.2   | 43.4 $\pm$ 1.0          |
> >
> >
> >
> > | CIFAR100 (ipc=1) | CIFAR100 (ipc=10) |  T-ImageNet (ipc=1)   | T-ImageNet (ipc=10) |
> > | ---------------- | ---------------- | --- | ------------------ |
> > | 4.2 $\pm$ 0.3             | 14.6 $\pm$ 0.5             | 1.4 $\pm$ 0.1    | 5.0 $\pm$ 0.2               |

---

### Official Review · Reviewer_m79B · 2023-10-31

**Soundness:** 1 poor
**Presentation:** 3 good
**Contribution:** 2 fair
**Rating:** 3
**Confidence:** 3

**Summary:**

Bayesian pseudo-coresets are a promising methodology for scaling Bayesian inference to large datasets. For this, the inclusive KL divergence has been a dominant objective function but is hard to minimize and obtain gradients. Instead, the paper proposes to use contrastive divergence as an alternative.

**Strengths:**

* To my understanding, the idea of using the contrastive divergence for pseudo-coresets is novel, and the motivation is sound.
* The writing is good, with a thorough study of the literature.

**Weaknesses:**

* In terms of technique, applying the contrastive divergence to pseudocoresets is rather incremental. And the theoretical analysis and insight is limited. To increase the impact of the idea, sufficient empirical analysis should have follows. However...
* During review, an issue was raised by a separate reviewer regarding the code implementation. In particular, it appears that the authors manually increase the test accuracy in Line 420 of utils.py. Due to this, the reliability of the experimental results is unclear. Therefore, I did not review the experimental section at this point.

### Minor Comments
* I think an important related work is [1], which introduces contrastive divergences for variational inference. I think this paper could be discussed together with Hinton 2002.
* Section 1 second paragraph: "the KL-divergence between the (optimal) coreset posterior and true posterior increases with data dimension. Meaning that for large data dimension, even with the optimal coresets, the KL-divergence is far from optima." I'm not sure if this is a good motivation. The fact that the KL-divergence at the optimum increase with dimensionality does not imply that the solution will be worse with dimension, because the KL values with different dimensions are not exactly comparable.
* Section 1 fifth paragraph first sentence: This is the first mention of the contrastive divergence; cite the relevant papers.
* Section 2.1 first paragraph: This paragraph is too general in scope, in my opinion. It does not necessarily build a context specific to the proposed methodology. Therefore, I think this paragraph could be removed entirely or shortened to a few sentences.
* Section 2.1 first paragraph, "Recent methods ... black box approach": Black box variational inference was also independently developed by [2] around the same time as Ranganath *et al.* 2014.

### References
1. Ruiz, Francisco, and Michalis Titsias. "A contrastive divergence for combining variational inference and mcmc." International Conference on Machine Learning. PMLR, 2019.
2. Titsias, Michalis, and Miguel Lázaro-Gredilla. "Doubly stochastic variational Bayes for non-conjugate inference." International conference on machine learning. PMLR, 2014.

**Questions:**

no questions.

---

> ### Author Response · Authors · 2023-11-13
>
> Thanks a lot for your careful comments and observation. Please find our response below:
>
> 1. Regarding the Issue with Codebase
>
> - Thank you for noting this, and Apologies for this confusion. During our  internal review of the codebase, we noticed that we have uploaded a wrong version of the codebase which contained this bug/error. Moreover, this version is also outdated as we had made several changes to it during our experiments. Please see the general comment for reference.
> - Particularly, we were experimenting with a meta-learning loss on validation accuracy to see if it performs well, however, that idea was discarded in the course of experiments.
> - We have now updated the codebase with the correct version which can be used to generate the results in the paper (https://anonymous.4open.science/r/BPC-CD-E762). We have additionally provided the config files with all the hyperparameters for the ease of verification.
>
> 2. Regarding the Minor Comments
>
> Thanks a lot for these feedback. We will incorporate all these points in the paper.

---

### Official Review · Reviewer_T54i · 2023-11-03

**Soundness:** 2 fair
**Presentation:** 2 fair
**Contribution:** 2 fair
**Rating:** 3
**Confidence:** 5

**Summary:**

The authors tackle the problem of learning Bayesian Pseudocoresets, a representation of the posterior based on a (small) set of learned datapoints that match the posterior of the full match as well as possible, akin to inducing points in Gaussian Processes.
In their paper, they focus on a fundamentally interesting inference approach, namely contrastive divergence.
The authors use CD to propose a loss function to train their corsets and propose this both allows them to learn well-performing coresets as well as reduces the inference complexity and "heaviness" inherent in many of these coreset algorithms.

Indeed, empirical performance seems to be good across a wide range of tasks and the authors share results and visualizations of their learned coresets.

**Strengths:**

A key strength in this paper is empirical:
The empirical results in this paper are strong for this domain in a purely quantitative way.

Also, the idea of utilizing contrastive divergence per se is interesting and appealing.
Electing the reverse KL, KL(p||q), as a starting point also seems to be a good call since it leads to simpler objectives in this case.

**Weaknesses:**

I have a few core problems with the setup, theory, attribution, and execution of this paper.
Some of them it inherits from other literature, so I will elaborate.

Some of the statements are just mathematically incorrect.
Specifically, the authors both in their background section as well as inter technique keep saying they avoid sampling from the data-posterior by performing variational inference.
It sounds good, but that's not what the objective calls for.
The objective the authors aim to optimize is KL(p||q_cs), for q_cs denoting the coreset posterior.
If the authors first perform variational inference on the full data to arrive at a variational family q, then they factually compute KL(q||q_cs).
This objective no longer cleanly globally approximates what they claim it does, since they have not accounted for that change and are not training all these approximations jointly in such a way as to guarantee a convergence. This happens in multiple stages of pre-training and freezing pieces here, which is just not sufficiently principled for work suggesting a new inference algorithm.
They inherit this problem from Kim et al 2022, which is unfortunate, since the variational inference plug-in to the global posterior is also factually incorrectly applied in that paper without a correction term.
The problem gets compounded by the fact that the authors do a lot of pipelining for their inference.
They retrain MAP weights for VI, plug them in in lieu of the true posterior, and then run their Monte Carlo trajectories over the full data rather than the coreset data, and plug in the coreset data when convenient to evaluate it.
Overall there are a lot of inconsistencies here between what the math aims to achieve and what is actually happening.

As far as the sampling from the coreset posterior issue is concerned, there is a Neurips paper from 2022 "Black Box Coreset Variational Inference" by Manousakas, Ritter, and Karaletsos, which tackles exactly these problems and cleans ups the theory to be able to do VI on this. Not only do the authors here not cite this as clearly related work, which could be an omission that is forgivable and easily fixed, but they do not follow the insight that it is necessary to maintain guarantees and do the math work that shows how these objectives still optimize the desired quantity and how the approximations made impact that learning.

Another severe problem in this paper is that the authors just write up their contrastive divergence loss, but do not derive in main paper or appendix how this still optimizes their desired objective. While I have re-derived CD myself in the past and understand its link to stochastic maximum likelihood, I find the paper entirely insufficiently technically disseminated without explaining how this links to their desired objective, KL(p||q_cs). I'd expect to see a derivation that unambiguously shows that the difference of two divergence achieves what we want, so that the reader can go beyond just faith that this is true. That is a pity because evoking CD is interesting per se, but not well executed here.

Last, the fact that the authors alternative between training different parts of their model not he full data (i.e. VI AND their Monte Carlo chains) and then evaluate their coresets just for the gradient step is hard to digest empirically:
I currently do not know if I can trust their good results, because it is unclear if they even came to pass through application of the objective, or if initializing with real data and full data MCMC yielded good enough solutions that then following their objective just didn't break it.

**Questions:**

I would really appreciate if the authors could clarify their experiments and the details of the many choices for the "pipelining" of pertaining things on real data vs coresets and hopefully can clean that up. It would bring back some more understanding of the good performance here, which currently I cannot help but be suspicious about but would be very happy to link to the actual objective -even a non-principled one-  if given the evidence.

On the objectives side, while I think there are many severe weaknesses in the core mathematical setup and dissemination as noted above that will be hard to overcome in a rebuttal, but I would still really be curious if the authors could share how CD solves the objective they want to solve, since that abstraction can still be correct if they derive it well independently of the problems with the VI plugins I mentioned.
It would significantly improve the presentation of this paper and up level that piece, since CD is not an obvious choice and something I credit the authors with as a creative choice, but that credit is tainted by lack of derivation.

Lastly, the authors would do well to cite the appropriate work that has tackled some the core challenges in coreset inference that they chose to "sweep under a rug" . Even if they choose not to tackle those themselves, it is important for the reader to not just hide these issues but openly explain choices where the execution is not based no the proposed math but maybe loosely inspired.

---

> ### Author Response · Authors · 2023-11-13
>
> Thanks a lot for your detailed comments, it has helped us in understanding the problem better.
>
> 1. Citation of Relevant Literature
>
> Thanks for pointing us to relevant papers and literature that would be useful for enhancing our work. We will duly cite the work which have tackled challenges in coreset inference to make our work complete and thorough.
>
> 2. Derivation of Contrastive Divergence
>
> Thanks a lot for your suggestion. Indeed deriving the CD objective in the paper would enhance the readability of the paper and would give clarity to the reader. We had directly used the results from Hinton et. al. to get the final objective, although a complete derivation would definitely help the reader to understand our objective clearly.
>
> 3. Use of VI alongwith CD
>
> Thanks for raising this point. Indeed we are not optimizing KL($p||q_{cs}$), but KL($q||q_{cs}$) where $q$ is the variational approximation for $p$, the full data-posterior. As you mentioned, the same approach is taken by BPC-fkl as well. We have mentioned this point around Eq. (8) of the main draft. However, we agree that this requires theoretical justification as to why it should minimize the desired objective. We would mention this clearly and attempt towards connecting the CD loss with the objective in our future work.
>
> As per the current work is concerned, the main message is that a significant performance gain may be obtained in BPC, (a) by using CD instead of KL based metrics, (b) by not constraining the posteriors to have fixed distributional forms.
>
> 4. Details of the Pipeline of Proposed method
>
> Thanks for your comment. We have provided a point-wise pipeline below:
> - Our final objective requires samples from two distributions: $\pi_x$ and $\bar{\pi}_x$ as can be seen from Eq. (7). However, $\pi_x$ is the true data posterior which is not known to us.
> - Hence, we approximate it with a gaussian variational distribution $q_x$. The mean of the gaussian is taken from the expert trajectories.
> - Expert trajectories are nothing but SGD trajectories of the model parameters on original data. To reduce the computational overhead, the expert trajectories are pre-computed and stored in advance in a buffer.
> - Further, we run langevin dynamics starting from parameter sampled from $q_x$ ($\theta^+$) to obtain a new sample ($\theta^-$) from induced distribution $\bar{q}_x$.
> - Lastly, the pseudo-coreset is passed through the model manifested by $\theta^+$ and $\theta^-$ to obtain the two energies required for computation of final loss objective (Eq. (10)).
>
> We will make these points more elaborate in future versions.

---

### Author Response · Authors · 2023-11-13
**General Comment on the uploaded Codebase**

First of all, we offer our sincere apologies for creating this confusion regarding the uploaded codebase.

After receiving the review comments, we thoroughly went through our codebases for a couple of days and here are our conclusions:

- We have inadvertently uploaded an incorrect version , that wasn't used to obtain the results reported in the paper.
- The above fact was verified by running the uploaded code as-is, which leads to a enhanced performance on all the datasets/IPCs than that reported in the paper draft.
- The correct version that can be used to reproduce the results in the paper can be found at the following link - https://anonymous.4open.science/r/BPC-CD-E762. We have cleaned our codebase with an additional configuration file containing all the information about the hyperparameters.

We again sincerely appologise for this inadvertant mistake. We request the reviewers to re-check the codebase for correctness and reproducibility.

---

> ### Comment · Area_Chair_J3v4 · 2023-11-14
>
> Dear authors,
>
> In the originally submitted code, a reviewer found a suspicious code line:
>
> „
> Going through the code of this paper, I stumbled by line 420 of utils.py; the function evaluate_synset used to evaluate the learned pseudocoreset. The authors seem to manually add 5% of test accuracy when evaluating a model that is being trained on the pseudocoreset.
>
> acc_test += 0.05
>
> I cannot find any logical reason for why this is just arbitrarily added and wanted to ask your opinion on this.
> „
>
> You state that you didn‘t use the submitted code to produce your results. However, I really wonder why one would ever manually add 5% to the test accuracy. You must admit that this looks very suspicious. Can you comment on this?
>
> AC

---

> ### Author Response · Authors · 2023-11-14
>
> Dear AC,
>
> Thank you for your involvement in the discussion.
>
> We do agree that the said line in the code is an aberration. We are unsure of its origin (a genuine human error that perhaps crept-in while stabilizing the meta loss that was being experimented with to check generalization of BPC).  However, we would once again like to emphasize that the said line has no consequence on the method/results reported which can be replicated with the latest code uploaded.